# QuestBench: Can LLMs ask the right question to acquire information in reasoning tasks?

**Belinda Z. Li**[*]
MIT CSAIL
bzl@csail.mit.edu

**Been Kim**
Google DeepMind
beenkim@google.com

**Zi Wang**
Google DeepMind
wangzi@google.com

## Abstract

Large language models (LLMs) have shown impressive performance on reasoning benchmarks like math and logic. While many works have largely assumed well-defined tasks, real-world queries are often *underspecified* and only solvable by acquiring missing information. We formalize this information-gathering problem as a constraint satisfaction problem (CSP) with missing variable assignments. Using a special case where only one necessary variable assignment is missing, we can evaluate an LLM's ability to identify the minimal necessary question to ask. We present QUESTBENCH, a set of underspecified reasoning tasks solvable by asking at most one question, which includes: (1) Logic-Q: logical reasoning tasks with one missing proposition, (2) Planning-Q: PDDL planning problems with partially-observed initial states, (3) GSM-Q: human-annotated grade school math problems with one unknown variable, and (4) GSME-Q: equation-based version of GSM-Q. The LLM must select the correct clarification question from multiple options. While current models excel at GSM-Q and GSME-Q, they achieve only 40-50% accuracy on Logic-Q and Planning-Q. Analysis shows that the ability to solve well-specified reasoning problems is not sufficient for success on our benchmark: models struggle to identify the right question even when they can solve the fully specified version. This highlights the need for specifically optimizing models' information acquisition capabilities.

## 1 Introduction

Large language models (LLMs) are increasingly being applied to reasoning tasks such as math [15, 34, 46], logic [70, 12, 17] and planning/coding [18, 59, 63, 6, 10]. These applications typically assume all necessary information has been provided.

However, real-world scenarios often violate this assumption. Users may omit crucial details in math problems, and robots might operate in environments with partial observability. In such cases, LLMs need the ability to proactively gather missing information, for instance, by asking clarifying questions.

This work investigates whether LLMs can identify and acquire the missing information neces-

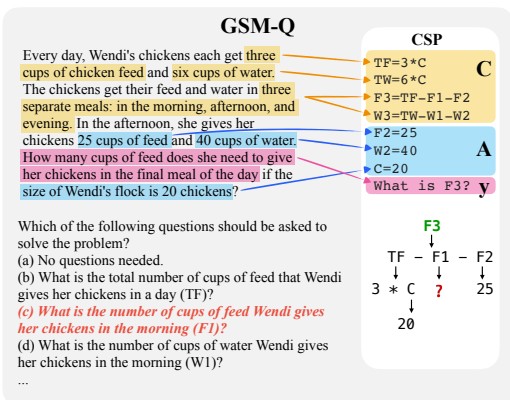

Figure 1: A multi-choice question-asking task in QUESTBENCH with *ground truth* answers for accuracy evaluation. We construct question choices using the CSP translated from the verbal problem.

---

[*]Work done as a student researcher at Google DeepMind. Code and dataset are available at https://github.com/google-deepmind/questbench. Author contributions in §A.

39th Conference on Neural Information Processing Systems (NeurIPS 2025) Track on Datasets and Benchmarks.

sary to solve reasoning tasks. We formalize this information-gathering problem as an underspecified Constraint Satisfaction Problem (CSP). Our key idea is that many reasoning tasks can be modeled as determining the value of a target variable given a set of variables and constraints. A problem is *underspecified* if and only if the value of the target variable cannot be inferred from the given information. This formalization helps pinpoint the difference between semantic ambiguity (where multiple valid interpretations exist, but each yields a solvable answer) and underspecification (where the problem is unsolvable without additional information). We address the latter in this work. Figure 1 shows an example of an underspecified CSP derived from a grade school math problem.

We focus on a simple special case of underspecified CSPs where we only need to acquire one unknown variable's value in order to solve for the target variable. We call these "1-sufficient CSPs" and construct QUESTBENCH: a benchmark of question asking problems based on 1-sufficient CSPs. The model is given an underspecified problem and multiple choices for which variable to ask about. It must pick a relevant variable, one whose *value isn't already known* (or derivable from the existing assigned variables), and is *sufficient* for computing the target variable.

QUESTBENCH has the following multi-choice problems, along with the correct solutions for each:

- **Logic-Q**: Logical reasoning tasks where the truth value of a missing proposition is needed to determine the correctness of a claim.
- **Planning-Q**: Blocks world planning problems in Planning Domain Definition Language (PDDL) [28], with partially observed initial states, where one additional observation is needed to disambiguate the shortest path to a goal.
- **GSM-Q/GSME-Q**: Grade-school math problems that are missing conditions needed to derive the solution. GSM-Q consists of verbalized forms of problems, while GSME-Q consists of equation forms of problems. Both are annotated by humans.

We benchmark state-of-the-art LLMs including GPT-4o [50], GPT-4-o1 Preview, Claude 3.5 Sonnet, Gemini 1.5 Pro and Gemini 2.0 Flash Thinking Experimental [25, 26] on QUESTBENCH. While these models demonstrate strong performance on GSM-Q and GSME-Q (over 80% accuracy), their performance on Logic-Q and Planning-Q is significantly lower (under 50% accuracy).

To investigate these discrepancies, we analyze the correlation between model accuracy and four "axes of difficulty" in QUESTBENCH (the number of variables in the problem, the number of constraints in the problem, the search depth required by a backwards search solution to the problem, and the expected number of guesses required by brute-force search solution to the problem), finding differing trends between domains. We also conduct ablation studies to assess LLMs' ability to 1) reason in the presence of sufficient information and 2) detect *whether* the problem is underspecified. We find that success on QUESTBENCH requires more than just the ability to reason with sufficient information.

Our contributions include (1) a constraint satisfaction-based problem formulation of asking clarification questions for reasoning, (2) QUESTBENCH, a novel benchmark with multi-choice problems for evaluating information-gathering abilities in reasoning tasks, (3) empirical evaluations and detailed analyses of SOTA or near-SOTA LLMs[2] on QUESTBENCH.

**Related work.** Most existing question-asking benchmarks focus on subjective or ambiguous tasks where multiple valid clarifying questions may exist, or the "right" clarification question is subjective [4, 20, 57, 62, 54, 8, 73, 45, 32]. This makes objective evaluation difficult. Our work focuses on reasoning tasks with clearly defined ground truth, where exactly one question is needed, allowing for reliable evaluation. More related work can be found in §B.

## 2 Problem formulation

Consider the following user request:

**Example 2.1.** *Please solve the math problem: Janet had some eggs (variable $x_0$) and ate one (variable $x_1$). How many eggs does she have now (target variable $y$)?*

The word problem can be parsed into equations $y = x_0 - x_1, x_1 = 1$. The LLM cannot compute target variable $y$ without knowing the value of variable $x_0$. Other examples can be found in Figures 1

---

[2]As of 03/2025.

to 3. In these cases, the desired behavior is for the LLM to ask the minimal set of questions that enables it to respond to the user query.

## 2.1 Distinguishing semantic ambiguity and underspecification

In this paper, we focus on problems that are semantically equivalent to CSPs,[3] which allow us to formalize underspecification. Solving a problem thus comprises of two stages:

1. A natural language query is parsed into a set of variables, a set of constraints, and a target variable corresponding to the desired response. In Example 2.1, there are variables $x_0$ (initial eggs), $x_1$ (eaten eggs), a constraint $y = x_0 - x_1$, and a target variable $y$ (current eggs) .
2. The model solves for the target variable in the CSP.

This formulation allows us to distinguish *underspecification* (missing variables assignments or constraints in the CSP at stage 2) from *semantic ambiguity* (multiple interpretations of a natural language query, inducing a distribution over possible CSPs at step 1). While prior work extensively addresses semantic ambiguity [42], this paper specifically focuses on underspecification, which is objectively evaluable using our CSP framework.

## 2.2 Formalizing underspecification

We formalize underspecification as follows. First, we define a CSP as a tuple $\langle X, D, C, A, y \rangle$:

- $X = \{x_i\}_{i=1}^N$ is a set of $N$ variables.
- $D = \{D_i\}_{i=1}^N$ is a set of domains, where $D_i$ is the domain of $x_i$.
- $C = \{c_j\}_{j=1}^M$ is a set of $M$ constraints. Each constraint $c_j$ can be viewed as a boolean function that takes in as input a list of variables in $X$, denoted as $[x_i]_{i \in I_j}$, and outputs True or False. We use the shorthand $c_j(\boldsymbol{x}_j)$ to denote $c_j([x_i]_{i \in I_j})$.
- $A = \wedge_{i \in I_A}(x_i = a_i)$ is a set of variable assignments, where each assignment is a pair $(x_i, v_i)$ with $v_i \in D_i$.
- $y$ is the target variable. Target $y$ must satisfy the constraints and the variable assignments, $\bigwedge_{j \in [M]} c_j(\boldsymbol{x}_j) \wedge A$.

**The *Known* predicate.** For convenience, we use $Known(x)$ to denote the value of variable $x$ is known. This can occur either if $x$ is directly assigned a value (i.e. $A \implies \wedge_{i \in I_A} Known(x_i)$), or $x$'s value can be derived from the existing assignments and constraints. In Example 2.1, once $x_0$ and $x_1$ are known, $y$ would also be known.

We can then formally define underspecified CSPs as follows.

**Definition 2.1.** *A CSP is* underspecified *if and only if* $\bigwedge_{j \in [M]} c_j(\boldsymbol{x}_j) \wedge A \implies \neg Known(y)$. *Otherwise it is* well-specified.

We use $Known(\mathcal{X})$ over a set of variables $\mathcal{X}$ to denote that the values of all variables in $\mathcal{X}$ are known, i.e., $Known(\mathcal{X}) = \bigwedge_{x \in \mathcal{X}} Known(x)$.

If knowing a set of variables is sufficient to determine the value of another variable, we call the set the "*sufficient set*":

**Definition 2.2.** *A set of variables* $\mathcal{X} \subseteq X$ *is a* sufficient set (for y) *if and only if*

$$\bigwedge_{j \in [M]} c_j(\boldsymbol{x}_j) \wedge A \wedge \text{Known}(\mathcal{X}) \implies \text{Known}(y).$$

For efficient question asking, we wish to find the smallest sufficient set for the target variable:

$$\hat{\mathcal{X}} = \arg\min_{\mathcal{X} \subseteq X} |\mathcal{X}| \quad \text{s.t. } \mathcal{X} \text{ is sufficient.}$$

In Example 2.1, a sufficient set for $y$ is $\{x_0\}$ since $Known(\{x_0\}) \implies Known(y)$.

Finally, we define the special case where the size of the smallest sufficient set is 1.

---

[3]This formulation encompasses queries where users have hard, deterministic preferences and expect a single correct answer. In general, we can think of user queries as a combination of (potentially probabilistic, fuzzy, or subjective) constraints and preferences.

**Definition 2.3.** *An underspecified CSP is a 1-sufficient CSP if the size of its smallest sufficient sets is 1. The variable in a smallest sufficient set is a sufficient variable of the problem.*

**Remarks.**  There could be multiple sufficient variables for a 1-sufficient CSP (and in the general case, there could be multiple, equivalently-sized smallest sufficient sets for any underspecified CSP). In QuestBench, choosing *any* sufficient variable is treated as correct.

### 2.3 Why 1-sufficient CSPs?

QuestBench is comprised of 1-sufficient CSPs. 1-sufficient CSPs are a fundamental building block towards $k$-sufficient CSPs. Performance on simpler 1-sufficient CSPs provides an approximate *upper bound* on a model's capacity for information acquisition, since any $k$-sufficient CSP can become 1-sufficient after $k - 1$ clarifying questions.

Another benefit of 1-sufficient CSPs is that they can be framed as multiple-choice problems which are easy to evaluate, by simply letting the model predict one choice. In contrast, $k$-sufficient problems may require questions to be asked sequentially, where the next question depends on the answer to the current question, making evaluation more complex. More discussions can be found in §5.

Finally, QuestBench includes significant variation in problem difficulty, even within the 1-sufficient setting, which we will describe in the next section.

### 2.4 Difficulty axes of 1-sufficient CSPs

Solving a 1-sufficient CSP entails identifying a sufficient variable for the target variable. In general, 1-sufficient CSPs can be solved by brute-force or backward search. Brute-force search checks the sufficiency of every variable one at a time. Backward search builds a search tree starting from the target variable (root node) and recursively searches all variables (child nodes) in each constraint (edges) involving the parent node. These two methods allow us to formalize the following properties associated with the difficulty of each 1-sufficient CSP:

- $|X|$: The total number of variables.
- $|C|$: The total number of constraints.
- $d$: The depth of the backward search tree.
- $\mathbb{E}_{\text{BF}}$: The expected number of random guesses until selecting a correct question.[4]

For example, in Example 2.1, there are $|X| = 3$ variables ($x_0, x_1, y$), $|C| = 1$ constraint ($y = x_0 - x_1$), and $d = 1$ search depth. Among choices *(a) No questions needed.*, *(b) What is the value of $x_0$?*, *(c) What is the value of $x_1$?*, only choice (b) is correct, so the expected number of guesses $\mathbb{E}_{\text{BF}} = 2$.

## 3 Dataset descriptions for QUESTBENCH

We construct 1-sufficient CSPs in logical reasoning (Logic-Q), planning (Planning-Q), and math (GSM-Q/GSME-Q) domains. We frame all problems as multiple choice questions for rigorous evaluation. Each problem instance is composed of a user request, the full set of question choices and a subset including correct questions. We evaluate whether models can pick out one of the correct questions from the question choices. See Table 5 for dataset statistics.

### 3.1 Logic-Q

SimpleLogic[5] [70] is a propositional logic benchmark, which consists of

1. a set of *rules* about an imaginary person named Alice, for example: *If Alice is jittery and strange, then Alice is stubborn.*
2. a set of *properties* that are true of Alice, for example: *Alice is smart. Alice is not stubborn.*
3. a question about an unknown property of Alice, for example: *Is Alice pleasant?*

---

[4]We assume the random guesses are made without replacement, so $\mathbb{E}_{\text{BF}} = \frac{|X|+1}{\mathfrak{s}+1}$, where $\mathfrak{s}$ is the number of sufficient variables.

[5]SimpleLogic has a MIT license. `https://github.com/joshuacnf/paradox-learning2reason`.

**Problem Definition.** We can define a CSP in this domain as follows:

- $X$ is a set of all the potential properties of Alice that appears in all rules (e.g. *strange, jittery, worried)*.
- $D = \{\{\text{TRUE}, \text{FALSE}\} \forall x \in X\}$. Each property in $X$ can be either be true or false.
- $A$ is the set of properties that we know to be true about Alice (e.g. *smart*, ¬*stubborn*).
- $C$ is the set of rules about Alice. $C$ takes the form of a conjunction of implicative constraints (e.g. in the above example, *strange* ∧ *jittery* $\implies$ *stubborn*).

$$C = \bigwedge_{i \in [M]} c_i$$

$$= \bigwedge_{i \in [M]} \left( \left( \bigwedge_{j \in [M_i]} x_{i,j} \right) \implies x_{i,M_i+1} \right)$$

where $x_{i,j}$ can take any value in the set $X$.

- $y$ is the property that we are being asked about, e.g. *pleasant* in the above example.

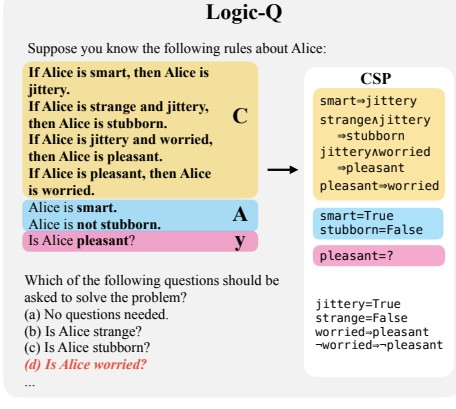

Figure 2: An example in Logic-Q. The prompt provided to the LM is on the left hand side. The ground truth answer is in *red*. The symbolic CSP used to construct the questions is shown on the right hand side.

To create Logic-Q, we discard the original, fully-specified variable assignments in the SimpleLogic dataset and construct new $A$'s such that a single additional property of Alice is sufficient to determine whether the target property is true or false. We do this by using backwards search to obtain 1) a set of all possible variable assignments that would imply $y$, and 2) another set for $\neg y$. We take the cross product between the sets and identify pairs which differ on a single variable assignment, meaning assigning that variable deterministically implies either $y$ or $\neg y$. See more details in Section C.1. In the end of this process, we get $\langle X, D, A, C, y \rangle$, which form a 1-sufficient CSP.

## 3.2 Planning-Q

Planning-Q is based on the Blocks World domain from PyperPlan[6] [1], a classic planning problem where the goal is to rearrange a set of blocks from an initial state to a goal state. Blocks World consists of **predicates** (e.g. `(ontable ?a)`), which can be applied to blocks (e.g. `c`) to construct an **atom** (e.g., `(ontable c)`). A **state** is a conjunction of atoms or their negations, e.g. `(ontable a)`, `(holding b)`, ¬`(on a b)`.

Not all states are valid; in a **valid state**, there are one or more actions that can be applied to it to transition to another state. Each action has a **precondition** specifying atoms the state must satisfy before applying the action, and an **effect** specifying atoms the next state must satisfy after applying the action. For example, `stack(?a, ?b)` has preconditions `(holding ?a)`, `(clear ?b)` and effect ¬`(holding ?a)`, ¬`(clear ?b)`, `(clear ?a)`, `(handempty)`, `(on ?a ?b)`.

Given an initial state and a goal state, the task is to plan the *shortest* action sequence to the goal state.

**Problem Definition** The problem can be expressed as a CSP $\langle X, D, C, A, y \rangle$ as follows:

- $X$ is the set of all atoms (predicates applied to objects) in the initial state.
- $D = \{\{\text{TRUE}, \text{FALSE}\} \forall x \in X\}$. Each atom must be either true or false.
- $A$ is the set of atoms that we know to be true or false in the initial state.

- $C$ is the set of constraints defining the action dynamics of the Blocks World domain. More formally, for any sequence of actions $[q_t]_{t \in [n]}$ and their corresponding sequence of states $[s_t]_{t \in [n]}$, each pair of current state $s_t$ and next state $s_{t+1}$ must satisfy the precondition and effect of action $q_t$, i.e.,

$$\forall p \in \text{PRE}(q_t), \ s_t \Rightarrow p \ \text{(preconditions of } q_t \text{ must hold in } s_t)$$

$$\forall e \in \text{EFFECTS}(q_t), \ s_{t+1} \Rightarrow e \ \text{(effects of } q_t \text{ must hold in } s_{t+1})$$

---

[6]PyperPlan has a GPL-3.0 license. `https://github.com/aibasel/pyperplan`.

- $y$ is the shortest action sequence from the initial state to the goal state.[7]

The Planning-Q dataset is a mix of 1-sufficient and well-specified problems, where at most 1 other atom is required to disambiguate the shortest action sequence to the target. In well-specified problems, the shortest action sequence is the same regardless of uncertainty about the initial state, in which case we expect the LLM to select the choice "*No questions needed*." We construct Planning-Q by deriving all possible initial states from which there is a single shortest path to the goal through backwards search, then removing up to one atom. More construction details can be found in Section C.2.

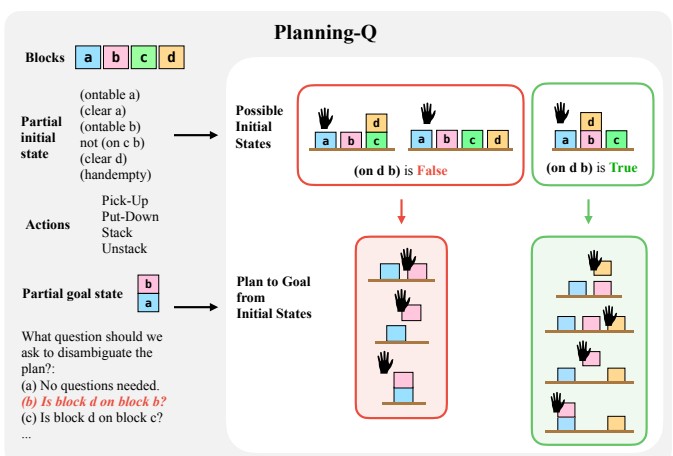

Figure 3: An example in Planning-Q. The ground-truth answer is given in *red*. The prompt given to the LM (left) includes the full task specification in PDDL, which we omit for simplicity and instead display visually. Possible initial states are constructed from the partial initial state and are grouped based on plans to the goal. These groups of initial states are used for constructing the questions.

## 3.3 GSM-Q and GSME-Q

In general, grade-school math problems can be parsed into simple algebra problems where a sequence of variable substitutions can fully solve the problem. We construct underspecified grade-school math problems from GSM-Plus' "distractor" setting[8] [46], which was derived from adding a single piece of distractor information to math problems in GSM8k [15] that is irrelevant to deriving the goal variable. This allows us to isolate whether LLMs can identify which specific variables are relevant to a goal variable. An example can be seen in Figure 1.

To construct GSM-Q/GSME-Q dataset out of GSM-Plus, we use human annotators to 1) check word problems for semantic ambiguity, and 2) translate each word problem into a CSP. Details can be found in Section C.3. To make these CSPs *underspecified*, we withhold both *distractor* variable assignments $\{d_i\}_{i=0}^n$ that *aren't* essential to computing $y$, as well as a single variable assignment $a \in A$ that *is* required for computing the goal, creating $\tilde{A} = A \setminus (\{d_i\}_{i=0}^n \cup \{a\})$. Asking about the value of the variable corresponding to $a$ is necessary and sufficient for deriving the value of target variable $y$.

These underspecified CSPs are consolidated into GSME-Q. To convert them back into word problems for GSM-Q, we show annotators the original word problem and the CSP form of the problem, and ask them to remove $a$ and the distractors $\{d_i\}_{i=0}^n$ from the word problem.

**Annotation Details** We recruited a total of 21 annotators (11 male, 10 female) from five countries to annotate our tasks. Annotators were all fluent in English and between the ages of $25 - 45$. We paid an average of $27 – $55 per hour per annotator.

# 4 Results and analyses

In this section, we present experiment results on QUESTBENCH. We evaluated GPT-4o [50], GPT-4-o1 Preview, Claude 3.5 Sonnet, Gemini 1.5 Pro, Gemini 1.5 Flash, Gemini 2.0 Flash Thinking Experimental [25, 26], and open-sourced Gemma models [27] in zero-shot (ZS), chain-of-thought (CoT), and four-shot settings (4S) settings. For GSM-Q and GSME-Q, we present results on a subset of 288 and 151 representative tasks, respectively, in order to focus our evaluation budget on more difficult domains. All results are based on publicly available APIs or models. Our experiments were done from 06/2024 to 03/2025, and some models/APIs may have been updated during this time.

---

[7]It is not meaningful to set $y$ to be *any* action sequence (not just the shortest one) to the goal, since all initial states can get to the goal state through some path in BlocksWorld, so no information needs to be gathered.

[8]GSM-Plus has a CC-BY-SA 4.0 license. https://huggingface.co/datasets/qintongli/GSM-Plus

|  |  | Logic-Q | Planning-Q | GSME-Q | GSM-Q |
|---|---|---|---|---|---|
| ZS | o1-preview | 49.91% | 42.31% | 98.01% | 91.32% |
|  | Gemini Flash Thinking 2.0 Exp 01-21 | 37.39% | 47.32% | 96.69% | 84.72% |
|  | Claude 3.5 Sonnet | 30.09% | 7.53% | 94.70% | 91.32% |
|  | GPT-4o | 27.39% | 24.49% | 96.69% | 86.81% |
|  | Gemini 1.5 Pro | 29.13% | 19.80% | 48.34% | 77.78% |
|  | Gemini 1.5 Flash | 16.78% | 8.52% | 96.69% | 71.88% |
|  | Gemma 2 2B IT | 0.09% | 10.76% | 32.45% | 49.65% |
|  | Gemma 2 9B IT | 26.00% | 14.36% | 75.50% | 40.97% |
|  | Gemma 2 27B IT | 29.39% | 7.64% | 73.51% | 71.18% |
| ZS + CoT | Gemini Flash Thinking 2.0 Exp 01-21 | 34.61% | 46.24% | 96.03% | 86.11% |
|  | Claude 3.5 Sonnet | 36.78% | 34.03% | 74.17% | 90.97% |
|  | GPT-4o | 31.57% | 16.36% | 99.34% | 92.01% |
|  | Gemini 1.5 Pro | 29.22% | 21.27% | 96.69% | 79.17% |
|  | Gemini 1.5 Flash | 18.70% | 9.29% | 98.68% | 76.74% |
|  | Gemma 2 2B IT | 0.26% | 3.01% | 56.95% | 68.40% |
|  | Gemma 2 9B IT | 24.61% | 10.27% | 56.95% | 36.46% |
|  | Gemma 2 27B IT | 31.22% | 9.96% | 80.79% | 79.51% |
| 4S | Gemini Flash Thinking 2.0 Exp 01-21 | 38.43% | 19.32% | 96.69% | 92.71% |
|  | Claude 3.5 Sonnet | 36.00% | 34.56% | 95.36% | 94.10% |
|  | GPT-4o | 25.57% | 11.48% | 96.03% | 84.03% |
|  | Gemini 1.5 Pro | 26.35% | 18.80% | 92.05% | 85.42% |
|  | Gemini 1.5 Flash | 15.48% | 18.29% | 95.36% | 80.90% |
|  | Gemma 2 2B IT | 25.13% | 7.51% | 90.73% | 37.50% |
|  | Gemma 2 9B IT | 21.57% | 5.15% | 86.75% | 55.56% |
|  | Gemma 2 27B IT | 26.26% | 10.71% | 96.69% | 66.67% |
| BFS | Depth 3 | 39.0% | 5.75% | 87.4% | 86.1% |
| BFS | Depth 5 | 80.6% | 27.2% | 100% | 100% |
| BFS | Depth 10 | 99.8% | 93.9% | 100% | 100% |

Table 1: Language model accuracies at predicting the right question in QUESTBENCH. As a baseline, we include breadth-first-search (BFS) up to a fixed depth on each dataset. Results show that the best models saturate GSME-Q and GSM-Q, but fail to perform beyond 50% on Logic-Q and Planning-Q.

## 4.1 Evaluation of clarification question accuracy

QUESTBENCH is a multi-choice benchmark with ground truth choices, each of which is a clarification question. We evaluated whether the LLM generates a *correct* choice for each question. Table 1 shows LLM accuracies in different setups. For reference, we also evaluated the accuracy of breadth-first-search up to a depth $n$.

Generally, all models we tested struggled to perform beyond $50\%$ on our Logic-Q and Planning-Q domains. Neither chain of thought nor few-shot examples resulted in significant gains across all models in either domain.[9] However, LLMs were generally able to saturate performance on GSME-Q and GSM-Q.[10] We suspect that this is due to these domains having a smaller number of variables and constraints, and requiring shallower search depth than the other two domains.[11] Additionally, LLMs may have been specifically studied and optimized for solving these sorts of math problems.

Comparing GSME-Q to GSM-Q, we found that models generally had a harder time reasoning about verbalized versions of problems than if they were presented in raw equation forms. We hypothesize that, at least for the problems in QUESTBENCH, identifying missing information requires building up a search tree, which can be easier if the problem were presented symbolically than verbally.

---

[9]Qualitative examples using Gemini 2.5 Pro, Google's newest public reasoning model as of October 2025, in these domains can be found in Section F.

[10]As a sanity check that these results hold for the entire GSME-Q and GSM-Q datasets, we run GPT-4o (ZS) on the full datasets and achieve 99.39% on GSME-Q and 84.05% on GSM-Q. Performance saturates on the full set, consistent with our findings on the smaller subset in the paper.

[11]We will present results in Table 2 and Fig. 6 that demonstrate performance is negatively correlated with number of variables, constraints, and search depth.

| | | ZS | | | | ZS + CoT | | | 4S | | |
|---|---|---|---|---|---|---|---|---|---|---|---|
| | | o1-preview | Gemini 2.0 FTE | Claude 3.5S | GPT-4o | Gemini 2.0 FTE | Claude 3.5S | GPT-4o | Gemini 2.0 FTE | Claude 3.5S | GPT-4o |
| Logic-Q | $d$ | **-0.23** | **-0.20** | **-0.19** | **-0.19** | **-0.15** | **-0.18** | **-0.22** | **-0.28** | **-0.28** | **-0.20** |
| | $\|X\|$ | **-0.27** | **-0.21** | **-0.15** | **-0.16** | **-0.11** | **-0.13** | **-0.22** | **-0.18** | **-0.27** | **-0.18** |
| | $\|C\|$ | **-0.22** | **-0.23** | **-0.11** | **-0.11** | **-0.22** | **-0.13** | **-0.18** | **-0.18** | **-0.24** | **-0.20** |
| | $\mathbb{E}_{BF}$ | **-0.41** | **-0.30** | **-0.26** | **-0.26** | **-0.16** | **-0.23** | **-0.35** | **-0.28** | **-0.31** | **-0.30** |
| Planning-Q | $d$ | 0.01 | **-0.11** | **0.07** | **0.04** | **-0.10** | **0.09** | **-0.06** | **-0.08** | **0.10** | **-0.06** |
| | $\|X\|$ | **-0.06** | **-0.11** | **-0.04** | **-0.07** | **-0.11** | -0.00 | **-0.09** | **-0.14** | -0.01 | **-0.10** |
| | $b$ | **-0.06** | **-0.11** | **-0.04** | **-0.07** | **-0.11** | -0.00 | **-0.09** | **-0.14** | -0.01 | **-0.10** |
| | $\mathbb{E}_{BF}$ | **-0.54** | **-0.14** | **-0.27** | **-0.28** | **-0.12** | **-0.56** | **-0.41** | **-0.35** | **-0.48** | **-0.41** |
| GSME-Q | $d$ | **-0.21** | **-0.26** | -0.14 | -0.10 | **-0.27** | -0.06 | -0.14 | **-0.20** | -0.05 | -0.10 |
| | $\|X\|$ | **-0.24** | **-0.27** | **-0.21** | -0.09 | **-0.31** | -0.11 | -0.14 | **-0.29** | **-0.25** | **-0.22** |
| | $\|C\|$ | **-0.24** | **-0.30** | **-0.24** | **-0.25** | **-0.34** | 0.04 | -0.14 | **-0.30** | **-0.26** | **-0.29** |
| | $\mathbb{E}_{BF}$ | **-0.24** | **-0.27** | **-0.21** | -0.09 | **-0.31** | -0.11 | -0.14 | **-0.29** | **-0.25** | **-0.22** |
| GSM-Q | $d$ | -0.03 | -0.01 | -0.06 | -0.03 | -0.01 | -0.09 | -0.01 | -0.00 | -0.01 | -0.03 |
| | $\|X\|$ | -0.02 | **-0.19** | **-0.13** | -0.09 | **-0.18** | **-0.19** | **-0.14** | -0.07 | -0.05 | -0.07 |
| | $\|C\|$ | 0.01 | -0.05 | -0.11 | -0.03 | -0.07 | **-0.16** | -0.09 | -0.01 | -0.05 | -0.05 |
| | $\mathbb{E}_{BF}$ | -0.02 | **-0.19** | **-0.13** | -0.09 | **-0.18** | **-0.19** | **-0.14** | -0.07 | -0.05 | -0.07 |

Table 2: Spearman's rank correlation coefficient between various axes and accuracy for predicting the right question. **Bolded** values indicate a statistical significance correlation between the axis and model accuracy ($p < 0.05$). Colored values are moderately strong correlations. We abbreviate Claude 3.5 Sonnet to Claude 3.5S and Gemini 2.0 Flash Thinking Experiment 01-21 to Gemini 2.0 FTE. We observe moderate to mild negative correlation between these factors and model performance in Logic-Q and Planning-Q, and sometimes in GSME-Q.

## 4.2 Correlation between problem complexity and clarification question accuracy

1-sufficient CSPs can generally be solved through either brute-force (forward) search or backwards search. We can approximately quantify the difficulty of each problem in QUESTBENCH based on the runtime of each search algorithm on that problem. If there is a correlation between the factors that determine search complexity and the performance of the LLM, this serves as a high-level signal for the types of mechanisms LLMs may be using to succeed in these tasks.

Specifically, we analyze the correlation between LLM performance and factors listed in Section 2.4. The runtime complexity of each search algorithm in the Logic-Q and Planning-Q domains can be expressed as a function of these factors, see Table 3, where the complexity of brute-force search holds with probability at least $1 - \delta$.

| | Logic-Q | Planning-Q |
|---|---|---|
| Brute-force Search | $O((\|X\| + \|C\|)\mathbb{E}_{\text{BF}}/\delta)$ | $O\left(4^{\|X\|}b^2\mathbb{E}_{\text{BF}}/\delta\right)$ |
| Backwards Search | $O\left(\|X\|^{\|C\|d}\right)$ | $O(b^{2d})$ or $O\left(3^{\|X\|d}\right)$ |

Table 3: Runtime complexities of brute-force and backwards search in Logic-Q and Planning-Q. Derivation details can be found in Sections E.1 and E.2.

We report Spearman's rank correlation coefficients between accuracy and these factors in Table 2, and plot accuracy across each factor in Figure 6. The runtime complexity derivations of each search algorithm in each domain are in Sections E.1 and E.2.

**Logic-Q** Table 2 and Figure 6 show that the LLMs' performances correlate negatively with backwards search depth $d$, number of variables $|X|$, number of constraints $|C|$, and expected number of brute-force guesses $\mathbb{E}_{BF}$. These correlations are statistically significant for most LLMs in this domain, indicating that in Logic-Q, brute-force and backwards search are moderately good approximations for the type of reasoning LLMs are required to perform.

|    |                  | Logic-Q |       |           | Planning-Q |       |           |
|----|------------------|---------|-------|-----------|------------|-------|-----------|
|    |                  | Acc     | Diff  | # samples | Acc        | Diff  | # samples |
|    | Gemini 2.0 FTE   | 37.4%   | 0.0%  | 1150      | 53.8%      | **6.5%** | 4273   |
|    | Claude 3.5 Sonnet| 30.8%   | **0.7%** | 1120   | 11.3%      | **3.8%** | 4368   |
|    | GPT-4o           | 27.4%   | 0.0%  | 1150      | 33.4%      | **8.9%** | 2275   |
| ZS | Gemini 1.5 Pro   | 29.6%   | **0.5%** | 1127   | 28.7%      | **8.9%** | 2175   |
|    | Gemini 1.5 Flash | 17.1%   | **0.3%** | 1127   | 16.1%      | **7.6%** | 1288   |

Table 4: Ablation: Language model accuracies at predicting the right question to ask on *only* filtered QUESTBENCH problems of which at least one well-specified version can be answered correctly by the model + prompt method. We use Gemini 2.0 FTE as a shorthand for Gemini Flash Thinking 2.0 Exp 01-21. See Section 4.3 for the experimental setup. **Acc** indicates accuracy on the filtered version of QUESTBENCH, **Diff** indicates increase in accuracy on the filtered vs. the original version, and **# samples** indicates the number of samples in the filtered dataset, out of 1150 and 7500 examples in the original Logic-Q and Planning-Q, respectively. We **bold** all accuracy differences greater than zero.

**Planning-Q** Results show that the LLMs' performances have very weak or no correlations with $d$, $|X|$, $b$, suggesting that LLMs may be using mechanisms other than the backwards search in planning. This is reasonable as these search algorithms generally require exponential-time search, while Transformers theoretically cannot perform non-polynomial-time-computation within a polynomial amount of chain-of-thought [47]. However, some models' accuracies have moderately strong correlation with $\mathbb{E}_{BF}$ despite nearly no correlation with $|X|$ or $b$. This indicates that those LLMs' behavior might be analogous to randomly guessing, or verifying one choice at a time (the $\mathbb{E}_{BF}$ factor in Table 3 for brute-force search), but perhaps not using breadth-first search for verification. Note that verifying one choice at a time is non-trivial, as it can entail rolling out entire trajectories – potentially from multiple possible initial states.

## 4.3 Is asking the right question harder than solving the problem?

To better understand the relationship between question-asking and reasoning, we conducted an ablation study filtering for examples where the LLM correctly answered the well-specified version of a task. By doing so, we can isolate the challenge of *asking* the right question from that of *solving* the underlying well-specified reasoning problem. This setup allows us to estimate how much more difficult question-asking is relative to reasoning with sufficient information.

We began by constructing a well-specified variant of QUESTBENCH, in which we restored one sufficient piece of missing information in each 1-sufficient problem. This yielded well-specified CSPs for which the target variable $y$ can be determined without gathering new information. We then prompted LLMs to directly solve these CSPs:

- In **Logic-Q**, the task is to determine the truth value of a target property about Alice.
- In **Planning-Q**, the model finds a valid plan from a given start state to a goal.
- In **GSM-Q**, we used the original GSM-plus math problems as-is.
- In **GSME-Q**, we used the GSM-plus problems translated into CSPs by the human annotators.

The prompts we used for this evaluation are provided in Section D.2, and evaluations on this well-specified variant of QUESTBENCH are reported in Table 6.

We then restricted our evaluation of question-asking to only those examples where the model correctly answered at least one corresponding well-specified variant. Results of this filtered evaluation are shown in Table 4, including accuracy, differences from unfiltered performance (Table 1), and the number of retained examples. Results for GSM-Q and GSME-Q can be found in Table 8.

We observed that question-asking accuracy increased slightly in **Planning-Q** after filtering, but remained effectively unchanged in **Logic-Q**, with neither achieving 100%. Notably, models reasoned significantly better on the well-specified version of Logic-Q than Planning-Q (see Table 6), yet this improvement did not translate to better question-asking performance in Logic-Q. This suggests that while reasoning ability for well-specified problems may be a necessary component of effective question-asking, it may not be sufficient for identifying what information is missing.

### 4.4 Detecting underspecification and recognizing uncertainty

We also investigated whether LLMs can *detect* underspecified problems and identify their own uncertainty. We did this by sampling the well-specified variants of QUESTBENCH from Section 4.3 and the original examples of QUESTBENCH, creating a mix of 1-sufficient and well-specified problems. For each problem, we asked LLMs to either 1) identify the target variable's value if the problem is well-specified (this is analogous to the setup in Section 4.3), or 2) say "not sure" if the problem is 1-sufficient. The prompts for this setting can be found in Section D.3.

Table 9 shows the F1 scores for the "*not sure*" class and results of Random, a baseline where a random set of examples (of the same size as ground-truth "*not sure*" set) are assigned "*not sure*". Surprisingly, we found that performance *did not always* improve substantially with model size, particularly in the Planning-Q domain where all models performed at most $5\%$ better than the Random baseline. We hypothesize that Planning-Q problems are sufficiently difficult that models cannot recognize uncertainty, and thus opt to guess randomly instead of answering (truthfully) that they are uncertain. As further support for this hypothesis, we found that Gemini Flash Thinking, Claude 3.5, and GPT-4o predicted "*not sure*" on only 1.4%, 4.6%, and 0.7% of cases in the zero-shot (no chain-of-thought) setting on Planning-Q, when the ground-truth ratio is 41.8%.

## 5 Discussion and Conclusion

We introduced QUESTBENCH for identifying underspecification and asking clarification questions in reasoning problems. We found that SoTA LLMs (as of 03/2025) are relatively good at identifying missing information in simple algebra problems, but struggle with complex tasks involving logic and planning. Their performance is sometimes negatively correlated with factors that increase problem complexity, such as search depth and expected number of brute-force guesses. Our findings suggest asking the right question requires more than the ability to reason with sufficient information.

When designing QuestBench, we made several design decisions for ease and rigor of evaluation, some of which limit the generalizability of our benchmark. Our framework serves as a first step towards formalizing underspecification. Below are some open problems that require future work.

**Constructing $k$-sufficient CSPs.** Our framework can be extended to problems with multiple missing variables. This can be done by holding out $k$ variable values, instead of just one, to construct $k$-sufficient CSPs. When doing so, it is crucial to consider whether the evaluation is for selecting all $k$ questions simultaneously (in a batch), selecting them sequentially (one by one), or a combination of the two: selecting batch by batch sequentially.

**Simulating users that answer questions.** Successfully handling complex, real-world tasks like medical diagnosis and intent elicitation often necessitates an iterative dialogue with multiple clarification questions. A major bottleneck in developing such systems is the challenge of automating the user's role for testing and evaluation. Building a simulator that can realistically answer the model's queries is a non-trivial task [31].

**Metrics of question-asking.** In practice, we might care about metrics beyond the correctness of clarification questions, for example, how easy the clarification questions are for the users to answer [32, 45, 30].

We believe the multiple-choice format for evaluating question-asking is a necessary precursor to open-ended formats for question asking, since it is valuable for automated and rigorous evaluation. However, once a new approach makes open-ended evaluation viable and reliable, our benchmark can be readily used by simply omitting the multiple-choice options.

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

# A  Author contributions

All authors contributed to brainstorming ideas and writing.

- Belinda Z. Li (bzl@csail.mit.edu): constructed and analyzed the dataset; implemented dataset construction and evaluation; human data collection; ran experiments and obtained results; analyzed empirical results.
- Been Kim (beenkim@google.com): advised project directions, dataset construction/analyses, human data collection and empirical results analyses.
- Zi Wang (wangzi@google.com): proposed and initiated project; constructed dataset prototypes; analyzed empirical results; advised dataset construction/analyses and human data collection.

# B  Related work

**Importance of Information Gathering.**   Actively gathering information is a critical skill for both humans and autonomous systems when solving problems under uncertainty. For humans, [14] demonstrates that asking questions plays a crucial role in cognitive development, helping children resolve ambiguity and learn more effectively. Similarly, AI/ML researchers have extensively explored strategies for sequential data acquisition in various domains, including concept learning [56, 3], active learning [16, 58, 36, 23, 55], Bayesian optimization [43, 44, 49, 5, 60, 35, 64, 24, 13, 21, 68, 22], reinforcement learning [39, 29, 61], and robot planning with partially observable states [40, 38, 52, 19] or unknown models [65–67]. [69] emphasizes the need for AI assistants to ask questions in the face of uncertainty, specifically in the context of code generation. Our work follows this tradition by evaluating how well large language models (LLMs) gather information through basic reasoning tasks.

**Ambiguity in user requests.**   Natural language queries often contain ambiguity for a variety of reasons. Prior work has examined ambiguity in the context of semantics [42], factual question-answering [48], task-oriented dialogue intents [54, 8, 73], personalized human preferences [45, 33, 11], and text-to-image generation [32]. [9] presents a visual question answering benchmark to identify epistemic and aleatory uncertainty, though the distinction between the two types of uncertainties can often be unclear. [72] introduces a taxonomy of ambiguity, categorizing it into issues like unfamiliarity and different semantic question types (e.g., "who," "what," "where").

In this paper, we focus on *underspecification* instead of ambiguity, where the user has not provided enough information for the LM to fulfill the request. This situation can arise because users may not know what information the model lacks, or what information is necessary to complete the task. We evaluate LLMs' ability to address underspecification in structured reasoning tasks.

**Information gathering benchmarks for LLMs.**   Most existing benchmarks focus on subjective or ambiguous tasks where there may be multiple valid clarifying questions, depending on context and user preference [4, 20, 7, 57, 62]. Task-oriented dialogue benchmarks [54, 8, 73] and preference elicitation tasks [45] involve inherently subjective problems where no universal "right" question exists. This makes objective evaluation of information-gathering abilities difficult in these settings. In contrast, our work focuses on reasoning tasks with a clearly defined ground truth. For each task, the model needs to ask exactly one question, allowing for reliable evaluation of LLMs' information-gathering capabilities.

**Question-asking methods for LLMs.**   Several methods have been proposed to enhance LLMs' ability to ask clarifying questions. These methods primarily address ambiguous or knowledge-based tasks, such as identifying a good recipe [2] or asking who won a sports event [71, 51]. Some approaches directly prompt LLMs to ask clarifying questions [45, 41], while others compute information gain to prioritize informative questions [53, 30, 33, 37]. [71] breaks down question-asking into three stages: detecting when clarification is needed, identifying the appropriate question, and responding based on new information. While these methods are promising, they primarily focus on subjective tasks or require substantial user simulation. Our work introduces a new setting that emphasizes generating accurate clarifying questions for underspecified reasoning tasks, where the correct question is objectively determinable.

## C  Dataset Construction Details

The dataset statistics can be found in Table 5.

| | #Tasks | $\lvert X \rvert$ / #Choices | $d$ | $\lvert C \rvert$ | $b$ | $\mathbb{E}_{\mathrm{BF}}$ |
|---|---|---|---|---|---|---|
| | | | | min – max (mean) | | |
| Logic-Q | 1150 | $10 - 59$ (37.9) | $1 - 12$ (4.1) | $8 - 108$ (47.8) | - | $1.2 - 15.0$ (6.1) |
| Planning-Q | 7500 | $25 - 64$ (46.0) | $1 - 14$ (6.9) | | $4 - 7$ (5.7) | $3.0 - 33.0$ (18.5) |
| GSM-Q | 23642 | $2 - 27$ (3.5) | $1 - 7$ (2.1) | $1 - 9$ (3.0) | - | $2.5 - 18.5$ (5.0) |
| GSME-Q | 6590 | $2 - 27$ (3.5) | $1 - 7$ (2.1) | $1 - 9$ (3.1) | - | $2.5 - 18.5$ (4.9) |

Table 5: Dataset sizes and properties (Section 2.4) for each domain in QUESTBENCH. We use $b$ to denote the number of blocks for the Blocks World planning problems.

### C.1  Logic-Q

We create 1-sufficient CSPs out of SimpleLogic problems by first discarding the $A$'s that are already present in the dataset. We then identify all assignments $A^{(y)} = \{A_i^{(y)} : A_i^{(y)} \implies y\}_{i=0}^{M}$ to (a subset of) variables in $X \backslash y$ which would imply $y$ is true, and similarly all assignments $A^{(\neg y)} = \{A_i^{(\neg y)} : A_i^{(\neg y)} \implies \neg y\}_{i=0}^{M'}$ that imply $y$ is false. These sets are found through recursive backwards search starting from $y$ or $\neg y$, see Section C.1.1.

Once we have the full set of assignments which imply $y$, to make them underspecified, we examine all pairs of assignments $(A_i^{(y)}, A_j^{(\neg y)}) \in \{A_i^{(y)}\}_{i=0}^{M} \times \{A_i^{(\neg y)}\}_{i=0}^{M'}$ where $A_i^{(y)}$ and $A_j^{(\neg y)}$ differ on an assignment to a single variable $x_d$. This means that if we remove $x_d$'s assignment from both $A_i^{(y)}$ and $A_j^{(\neg y)}$ (creating $A_i^{(y)} \backslash x_d$ and $A_j^{(\neg y)} \backslash x_d$), then knowing $\left( (A_i^{(y)} \backslash x_d) \wedge (A_j^{(\neg y)} \backslash x_d) \right)$ is true means knowing $x_d$'s value is sufficient to determine whether $y$ or $\neg y$ is true,

$$Known(x_d) \wedge (A_i^{(y)} \backslash x_d) \wedge (A_j^{(\neg y)} \backslash x_d) \implies Known(y).$$

We conduct further checks (see Section C.1.2) to ensure that the assignments themselves do not already imply a value for $y$,

$$(A_i^{(y)} \backslash x_d) \wedge (A_j^{(\neg y)} \backslash x_d) \implies \neg Known(y),$$

discarding any assignment from that doesn't satisfy the above property. We define $\tilde{\mathcal{A}}$ as the set of assignments satisfying the two properties above, and $\tilde{A}$ as an element of this set.

$$
\begin{aligned}
\tilde{\mathcal{A}} = \big\{ (A_i^{(y)} \backslash x_d) &\wedge (A_j^{(\neg y)} \backslash x_d) : \\
&\exists x_d \in X, \\
&\left( Known(x_d) \wedge (A_i^{(y)} \backslash x_d) \wedge (A_j^{(\neg y)} \backslash x_d) \implies Known(y) \right) \\
&\wedge \left( (A_i^{(y)} \backslash x_d) \wedge (A_j^{(\neg y)} \backslash x_d) \implies \neg Known(y) \right) \big\}
\end{aligned}
\tag{1}
$$

where the sufficient set of each $\tilde{A} \in \tilde{\mathcal{A}}$ is defined as

$$
\mathcal{C}(\tilde{A}) = \left\{ x : \left( \mathrm{Known}(x_d) \wedge \tilde{A} \implies \mathrm{Known}(y) \right) \wedge \left( \tilde{A} \implies \neg \mathrm{Known}(y) \right) \right\}.
\tag{2}
$$

#### C.1.1  Constructing all assignments that imply goal is true

We begin by doing backwards search through the constraints $C$ to iteratively derive the set of assignments which imply $y$ is true. At each iteration, we keep track of a disjunction of conjunctions of variables that must be set in order for the goal variable to be true. We start from the most trivial assignment – just setting goal variable itself $y$ to true.

$$g_0 = y$$

We then find all rules which imply $y$ is true and add the relevant variable assignments to our set. For example, suppose we have rule $a \wedge b \rightarrow y$ and $c \wedge \neg d \rightarrow y$, then at the first iteration, we expand our disjunction of conjunctions to include

$$g_1 = (a \wedge b) \vee (c \wedge \neg d)$$

We then iterate through each conjunction, and apply the above procedure on each variable in the conjunction. For example, if we have $e \wedge f \implies b$ and $e \wedge g \wedge h \implies b$ and $e \wedge x \implies c$ in the constraints, then we expand as:

$$g_2 = ((a \wedge ((e \wedge f) \vee (e \wedge g \wedge h)))) \vee ((e \wedge x) \wedge \neg d)$$

Which we can then apply the distributive property to, obtaining

$$g_2 = (a \wedge e \wedge f) \vee (a \wedge e \wedge g \wedge h) \vee (e \wedge x \wedge \neg d)$$

More formally, we decompose each constraint $C_i$ into $\text{PREMISES}(C_i) \implies \text{CONCLUSION}(C_i)^{12}$, where $\text{PREMISES}(C_i)$ is a set of terms that participate in the conjunction, while $\text{CONCLUSION}(C_i)$ is a single term implied by $\text{PREMISES}(C_i)$. Backwards search is thus formalized as follows:

$$g_0 = y$$

$$g_1 = \bigvee_{i}^{[N]} \left( \bigwedge_{j}^{[M_i]} t_{i,j} \right), \ \forall t_{i,j} \in \text{PREMISES}(C_i), \forall C_i \text{ where } \text{CONCLUSION}(C_i) = y$$

$$g_2 = \bigvee_{i}^{[N]} \left( \bigwedge_{j}^{[M_i]} \left( \bigvee_{k}^{[N_j]} \left( \bigwedge_{\ell}^{[M_k]} t_{i,j,k,\ell} \right) \right) \right), \ \forall t_{i,j,k,\ell} \in \text{PREMISES}(C_{i,j,k}),$$

$$\forall C_{i,j,k} \text{ where } \text{CONCLUSION}(C_{i,j,k}) = t_{i,j}, \cdots$$

$$= \bigvee_{i}^{[N]} \bigvee_{\substack{k', \\ \forall (j,k') \in ((0,k'),\cdots,(M_i,k')) \\ \forall ((0,k'),\cdots,(M_i,k')) \in \\ \bigtimes \{\{(j,k) \forall k \in [N_j]\} \forall j \in [M_i]\}}} \bigwedge_{j}^{[M_i]} \bigwedge_{\ell}^{[M_k]} t_{i,(j,k'),\ell}$$

$$\forall t_{i,(j,k'),\ell} \in \text{PREMISES}(C_{i,(j,k')}),$$
$$\forall C_{i,(j,k')} \in \bigtimes_j \{C_{i,j,k} \forall C_{i,j,k} \text{ where } \text{CONCLUSION}(C_{i,j,k}) = t_{i,j}\}$$
taking all combinations of $k$ rules that can form each $j$ term

$$= \bigvee_{i_2} \bigwedge_{j_2} t_{i_2,j_2} \qquad \text{re-indexing}$$

$$\cdots$$

to infer all sets of variable assignments that implies $y$. Similarly, we repeat the process starting from $\neg y$.

This gives us the full set of variable assignments $A^{(y)}$ which imply $y$ is true. We also repeat this backwards-search procedure starting from $\neg y$ to get the full set of variable assignments $A^{(\neg y)}$ which imply $\neg y$ is true.

### C.1.2 Checking 1-sufficiency

After constructing potential 1-sufficient assignments $A^{y,-1} = \{(A_i^{(y)} \backslash x_d) \wedge (A_j^{(\neg y)} \backslash x_d) \forall x_d \in X\}$, we conduct several further checks to ensure they are 1-sufficient:

---

[12] Note that any rule of form $a \wedge b \wedge c \implies d$ is equivalent to $a \wedge b \wedge \neg d \implies \neg c$, $a \wedge \neg b \wedge c \implies \neg d$, etc. We consider all possible cycles by writing $C_i$ in the form of a disjunction, $\neg a \vee \neg b \vee \neg c \vee d$ and seeking all disjunctions that contain a particular term (e.g. $d$), meaning they are implied by the conjunction of the negation of the rest of the terms (e.g. $a \wedge b \wedge c$).

1. First, we check that
$$\forall A_i^{y,-1} \in A^{y,-1},$$
$$\left(\forall A_i^{(y)} \in A^{(y)}, A_i^{y,-1} \implies\!\!\!\!\!/\ \ A^{(y)}\right) \wedge$$
$$\left(\forall A_i^{(\neg y)} \in A^{(\neg y)}, A_i^{y,-1} \implies\!\!\!\!\!/\ \ A^{(\neg y)}\right)$$

   This ensures that $A_i^{y,-1}$ by itself is not sufficient to determine the value of $y$. We throw away any $A_i^{y,-1}$ that does not satisfy these two constraints.

2. For any $A_i^{y,-1} \in A^{y,-1}$, if exists another $A_{i'}^{y,-1} \in A^{y,-1}$ such that $A_{i'}^{y,-1} \implies A_i^{y,-1}$, then all variables in the sufficiency set of $A_{i'}^{y,-1}$ is also in the sufficiency set of $A_i^{y,-1}$. However, resolving $A_{i'}^{y,-1}$ may require shallower backwards search than resolving $A_i^{y,-1}$ (see Section E.1). To ensure that the LM conducts search to the full depth required for resolving $A_i^{y,-1}$, we construct an "*invalid set*" consisting of the elements of the resolution sets of $A_{i'}^{y,-1}$. During test-time, we tell the LM that it cannot ask about the value of any $x_d'$ in the sufficiency set of $A_{i'}^{y,-1}$.

## C.2 Planning-Q

We introduce some notation to describe the dataset construction procedure for Planning-Q. The fully-specified version of the Blocks World task with $n$ blocks can by characterized as $\langle X, S, Q, s_0, y \rangle$ where

- $X$ is the full set of atoms $p$ that can be true of a state.
- $S$ is the set of *physically-possible* fully-specified states, which can be represented as a full set of assignments from all variables in $X$ to $\{\text{TRUE}, \text{FALSE}\}$.
- $Q$ is a set of actions that operate on the current state and transitions it to a next state. Each action $q \in Q$ has a set of preconditions which must hold for the action to be applicable in the state, and a set of effects which hold after the action is applied. Pre-conditions and effects can be expressed as a conjunction of atoms $x$ or negated atoms $\neg x$ for any $x \in X$. In the blocks setting, there are 4 types of actions which can be enacted on each block:

```
pick-up(?x)
    :precondition (and (clear ?x) (ontable ?x) (handempty))
    :effect (and (not (ontable ?x)) (not (clear ?x))
                 (not (handempty)) (holding ?x))
put-down(?x)
    :precondition (holding ?x)
    :effect (and (not (holding ?x)) (clear ?x)
                 (handempty) (ontable ?x))
stack(?x, ?y)
    :precondition (and (holding ?x) (clear ?y))
    :effect (and (not (holding ?x)) (not (clear ?y))
                 (clear ?x) (handempty) (on ?x ?y))
unstack(?x, ?y)
    :precondition (and (on ?x ?y) (clear ?x) (handempty))
    :effect (and (holding ?x) (clear ?y) (not (clear ?x))
                 (not (handempty)) (not (on ?x ?y))))
```

- $s_0$ is the initial state, which is in $S$.
- $y$ is a conjunction of goal propositions which we wish to be true at the end of a plan. There are 6 possible $y$'s in this dataset, which were written manually by the authors:

```
{(on b a),
 (and (on b a) (on c b)),
 (and (on b a) (on d c)),
 (and (on b a) (ontable a)),
 (and (on b a) (ontable a) (on c b)),
 (and (on b a) (ontable a) (on d c) (ontable c))}
```

Given the above, models are expected to construct an **optimal action sequence** $\hat{\tau} = [q_0, q_1, \cdots, q_{k-1}]$ where $q_t \in Q$ for all $q_t$ in the trajectory.

**Definition: (Optimal) Action Sequence.** An action sequence $\tau$ enables the robot to go from initial state $s_0$ to a goal state $s_k$ where the goal conditions are satisfied ($s_k \implies y$). We use TRANSITION($s_0, \tau$) to denote the state of the robot after taking action sequence $\tau$ from state $s_0$. We also use $s_0, \cdots s_k$ to denote the sequence of intermediate states the robot goes through when taking the action sequence, where action $a_i$ results in state $s_{i+1}$. An optimal sequence $\hat{\tau}$ is the shortest path that satisfy TRANSITION($s_0, \tau$) implies $y$.

$$\hat{\tau} = \arg\min_{\tau} |\tau| \quad \text{s.t. TRANSITION}(s_0, \tau) \implies y \tag{3}$$

**Definition: Planner.** We use $\Psi_y$ to denote an optimal planner that can map initial states $s_0$ to an optimal action sequence $\hat{\tau}$ to the goal $y$. The planner is implemented through breadth-first-search, on a search graph where the nodes are the states and the edges are the actions, starting from state $s_0$ and terminating in a state where $y$ is true.

In order to make the planning problem underspecified, we construct a version of it where certain literals (a literal is an atom or its negation) in $s_0$ are withheld, such that the initial state is only partially observed.

**Definition: Partial state.** A partial state is one where a subset of atoms (variables in $X$) have been assigned values.

**Definition: Consistent set.** We say a full state $s$ is consistent with a partial state $\tilde{s}$ if $s \in S$ and $s \implies \tilde{s}$. The set of all states which are consistent with $\tilde{s}$ is called the consistent set for $\tilde{s}$, which we denote with $F(s)$.

We begin by inferring all partial states $\tilde{s}_0$ where there is only a single possible optimal action sequence from any $s_0 \in F(\tilde{s}_0)$ to the goal. In other words, if we know $\tilde{s}_0$, we know the optimal trajectory to the goal. This gives us all the well-specified CSPs.

$$\tilde{\mathcal{S}}_0 = \{\tilde{s}_0 : \exists\hat{\tau}, s_0 \in F(\tilde{s}_0) \implies \Psi(s_0) = \hat{\tau}\} = \{\tilde{s} : \tilde{s} \implies \text{Known}(\tau)\}. \tag{4}$$

The construction process for this set is given in Section C.2.1.

In order to construct 1-sufficient initial states, we withhold a single atom from each $\tilde{s}_0 \in \tilde{\mathcal{S}}_0$ and check that knowing the truth value of a single additional atom in $X$ is *necessary and sufficient* in order to disambiguate **a single optimal action sequence** to the goal. Details are given in Section C.2.2.

$$\tilde{\mathcal{S}}_0' = \left\{\tilde{s}_0' : \exists x \in X, \big(\text{Known}(x) \wedge \tilde{s}_0' \implies \text{Known}(\tau)\big) \wedge \big(\tilde{s}_0' \implies \neg\text{Known}(\tau)\big)\right\} \tag{5}$$

where the sufficient set of each $\tilde{s}_0'$ is defined as

$$\mathcal{C}(\tilde{s}_0') = \{x : \big(\text{Known}(x) \wedge \tilde{s}_0' \implies \text{Known}(\tau)\big) \wedge \big(\tilde{s}_0' \implies \neg\text{Known}(\tau)\big)\}. \tag{6}$$

### C.2.1 Deriving all well-specified partial states with the same optimal action sequence to goal

Given goal conditions $y$, first, we perform backwards breadth-first-search from $y$ to derive the full set of optimal partial-state trajectories that end at $y$.

**Definition: Partial-state Trajectories.** A trajectory $\tilde{\tau}$ where the intermediary states are partial states, e.g. $[\tilde{s}_0, q_0, \tilde{s}_1, q_1, \cdots, \tilde{s}_k]$, where $\tilde{s}_{1\cdots K}$ are partial states. Partial-state trajectories are valid if for any consecutive state sequence $\tilde{s}_t, q_t, \tilde{s}_{t+1} \in \tilde{\tau}$, applying $q_t$ to any full state $s_t \in F(\tilde{s}_t)$ arrives at some $s_{t+1} \in F(\tilde{s}_{t+1})$. They are optimal if the trajectory is the shortest trajectory from $\tilde{s}_0$ to $\tilde{s}_k$.

Starting from the atoms in $y$, we iteratively search each action and deduce the partial states from which applying that action result in $y$.

$g_0 = \{[y]\}$

$g_1 = \{[\tilde{s}, q, y], \ \forall q \in Q \ \forall \tilde{s} \text{ s.t. TRANSITION}(\tilde{s}, q) \implies y\}$

    *find all partial states $\tilde{s}$ that transition into $y$ on some action $q$,*

    *prepend $\tilde{s}, q$ to the existing trajectories*

    $\cdots$

$g_{i+1} = \{[\tilde{s}, q, \tilde{\tau}], \ \forall q \in Q \ \forall \tilde{s} \text{ s.t. } \exists \tilde{\tau} \in g_i \text{ where TRANSITION}(\tilde{s}, q) \implies \tilde{\tau}[0]\}$

    *find all partial states $\tilde{s}$ that transitions from some action $q$ into an initial state $\tilde{\tau}[0]$*

    *of a trajectory $\tilde{\tau}$ found in the prior iteration $g_i$. Prepend $\tilde{s}, q$ to that trajectory*

    $\cdots$

We expand a search tree where each branch of the tree is partial trajectory. We terminate search for that branch if we arrive at a partial state $\tilde{s}$ that is implied by a partial state we have already encountered (meaning we have already considered that partial state). This means in the worst case, we exhaust the space of all possible partial states. When all branches have terminated, we have the set of all optimal partial-state trajectories to the goal $y$, by taking the union $g_0 \cup g_1 \cup \cdots$. The set of well-specified partial states $\tilde{\mathcal{S}}_0$ is the initial states of all these trajectories.

### C.2.2 Creating 1-sufficient partial states

To create 1-sufficient partial states from $\tilde{\mathcal{S}}_0$, we remove one proposition $x_d$ from each $\tilde{s}_0 \in \tilde{\mathcal{S}}_0$ to create $\tilde{\mathcal{S}}_0' = \{\tilde{s}_0 \backslash x_d \forall x_d \in \tilde{s}_0 \forall \tilde{s}_0 \in \tilde{\mathcal{S}}_0\}$ where $\tilde{s}_0' = \tilde{s}_0 \backslash x_d$ for some $\tilde{s}_0$, and check that each of the following holds:

1. For all other $\tilde{s}_0'' \in \tilde{\mathcal{S}}_0$, we check whether $\exists x \in X, \tilde{s}_0'' = \tilde{s}_0' \wedge x$, meaning that a single additional true atom brings us to another 1-sufficient partial state. In this case, we assign the salient atom $x$ where $\tilde{s}_0'' = \tilde{s}_0' \wedge x$ to FALSE, to eliminate the possibility more than one question must be asked: for example, if $x$ is asked and turns out to be true, then we are brought to $\tilde{s}_0''$, which we know is 1-sufficient, meaning we must ask at least one other question.
2. For all physically-valid, well-specified states $s_0' \in F(\tilde{s}_0')$ consistent with the 1-sufficient state $\tilde{s}_0'$, we check that there are at most two possible distinct optimal action sequences to the goal condition $y$:
$$\left| \{\Psi_y(s_0'), \ \forall s_0' \in F(\tilde{s}_0')\} \right| \leq 2$$
   (a) If there is 1 unique action sequence, then we expect the LM response to be "*No questions needed.*"
   (b) If there are 2 unique action sequences $\tau_1, \tau_2$, then we separated out $\tilde{S}_0$ into $\tilde{S}_0^{(1)}$ and $\tilde{S}_0^{(2)}$, where the optimal action sequence from all states in $\tilde{S}_0^{(1)}$ to the goal is $\tau_1$, while the optimal action sequence from all states in $\tilde{S}_0^{(1)}$ to the goal is $\tau_2$ ($\tilde{S}_0 = \tilde{S}_0^{(1)} \cup \tilde{S}_0^{(2)}$). We find the set of differentiating attributes between $\tilde{S}_0^{(1)}$ and $\tilde{S}_0^{(2)}$, that is to say, all attributes of states in $\tilde{S}_0^{(1)}$ that aren't present in any state of $\tilde{S}_0^{(2)}$, or vice versa (all attributes of states in $\tilde{S}_0^{(2)}$ that aren't present in any state of $\tilde{S}_0^{(1)}$). If any of these questions are asked, they would disambiguate whether the optimal action sequence is $\tau_1$ or $\tau_2$.

This ensures that for all $\tilde{s}_0'$, asking about the truth value of $x_d$ fully determines a unique optimal trajectory from the state to the goal conditions. If $\tilde{s}_0'$ passes all of the above checks, we add it to the set of 1-sufficient partial states $\tilde{\mathcal{S}}_0'$

### C.3 GSME-Q

First, we ask annotators to try and solve the word problems on their own. If they cannot solve the problem due to semantic ambiguity, or answer the problem in a way that doesn't match the original answer in GSM-Plus (due to interpreting the problem differently, or erroneous problems in GSM-Plus), we discard the problem entirely.

Next, we ask annotators to try and parse each math word problem into a set of variables $X$, equations $C$, assignments $A$, and a goal variable $y$. We assume domains $D$ for each variable is the set of natural numbers. We ask three different annotators provide CSPs for each math problem, resulting in different CSPs (different annotators may have different interpretations of a problem). We further perform automated checks to ensure the annotated CSPs actually result in the correct answer, discarding any CSP which we weren't able to parse or provided incorrect answers. We use all valid CSPs resulting from this process, including different CSPs corresponding to the same math problem. This gives us the set of fully-specified, high quality CSPs.

Full instructions we provided to annotators can be found below with interface screenshot in Figure 4.

> You will be presented with a series of math problems. These math problems are written in words and may be ambiguous. Your task is to try to solve the problem. The problem may be ambiguous,

which would make it unsolvable. However, if the problem is solvable, you will be asked to provide the answer, and may additionally be asked to translate the problem into a set of variables and equations given the information present in the problem. Two examples are provided below. Please read carefully and make sure you understand before proceeding.

Math problem 1:
*If there are 10 eggs in a basket. Alice buys more eggs and increases her egg quantity by 200%, but she had also sold half of her eggs by then. How many eggs are there total?*

You will be asked to try and solve the problem to check if it is ambiguous.
1. Try to solve this problem. What is the answer?: [text box]
   ☐ Not sure. Explain why: [text box E]
      What questions, when answered, could clarify this problem?: [text box Q]
In this case, the problem is ambiguous. You should check off "Not sure" and write why the problem is ambiguous in the explanation text box E. For example, in this case, you may write: it is unclear whether "increases by 200%" means 200% or 300% of her original amount. Furthermore, it is unclear which came first: did she sell half her eggs before increasing by 200%, or did she buy 200% more eggs first, then sell half her eggs.

Next, you should write some questions that could be asked to clarify this problem in text box Q. For example, you may write "does an increase by 200% mean 200% or 300% of the original amount?", "which happened first, Alice buying more eggs or Alice selling half her eggs?"

Here are some other examples of ambiguous questions that raters have found in this dataset. **Note**: there may be some subjectivity when deciding whether or not a particular problem is ambiguous. Please base it off your own interpretation.

| Problem | Explanation |
|---|---|
| Janet buys a brooch for her daughter. She pays $500 for the material to make it and then another $800 for the jeweler to construct it. After that, she pays 10% of that to get it insured. How much did she pay? | The antecedent of "that" in "10% of that" is unspecified. |
| Josh decides to try flipping a house. He buys a house for $80,000 and then puts in $50,000 in repairs. This increased the value of the house by 150%. How much did he make? | What should be considered the initial value of the house is unclear. It could be taken as the initial purchase price or the initial purchase price plus repairs. Furthermore, it is unclear whether "increase by 150%" means 150% or 250% or the price. |
| Jason has a phone plan of 1000 minutes per month. Every day he has a 15-minute call with his boss, and he's had 300 extra minutes of call this month to other people. How many minutes does Jason have left if this month has 30 days? | The day of the month is not specified. |
| In a 60-item quiz, 40% of the questions are easy, and the rest are equally divided as average and difficult questions. If Aries is sure to get 75% of the easy questions, and half of the average and difficult questions correctly, how many points is she sure to get? | The number of points per question is not specified. They could all be worth one point or they could be weighted differently. |
| Mara added 3 slices of cake to a plate that already had 2 slices on it. She was getting hungrier so she tripled the number of slices she currently has. She ate 2 slices and while she was distracted, her friend stole 5 slices off her plate. What number of cake slices remained on the plate? | The order of events is unclear. Because of the temporal mismatch between "was getting" and "currently has" in the same sentence, a reader cannot know whether "the number of slices she currently has" refers to the number of slices before or after adding the 3. |

**Note 2**: the problem may be ambiguous in more ways than one. Please explain all ways the problem is ambiguous.

Math problem 2:
*If there are 10 eggs and 2 in a basket, and there are twice as many eggs in a second basket, how many eggs are there total?*
    1. Try to solve this problem. What is the answer?: [text box]
        ☐ Not sure. Explain why: [text box E]
            What questions, when answered, could clarify this problem?: [text box Q]
In this case, you should answer 30. This is because there are 10 eggs in the first basket + 20 in the second basket, so 30 total. You should not check off "Not sure."

In cases where you did not check off "Not sure", you may be additionally asked to translate the problem into a series of equations, together with the variables that appear in the equations and the goal of the problem.
    2. Please translate the above math problem into a list of variables, a list of equations, and a goal variable. Ensure that your translation is equivalent to the variables above.
For example, in this case, the problem may be translated as follows:

    Variables:
    A = 10 [Number of eggs in the first basket]
    B [Number of eggs in the second basket]
    T [Total number of eggs]

Equations:
B = 2 * A [There are twice as many eggs in the second basket as the first.]
T = A + B [The total number of eggs is the sum of the eggs in the first and second baskets.]

Goal: T. How many eggs are there total?

If the value of a variable is directly mentioned in the math problem, you should write down the value. For example, A = 10. If the value of a variable is not directly mentioned in the math problem, you should not write down the value, even if it can be computed. Here are some examples of incorrect translation:
INCORRECT EXAMPLE 1

Variables:
A = 10 [Number of eggs in the first basket]
T [Total number of eggs]

Equations:
T = A + B. The total number of eggs is the sum of the eggs in the first and second baskets.

Goal: T. How many eggs are there total?

The above example is missing a variable (the number of eggs in the second basket) and an equation.
INCORRECT EXAMPLE 2

Variables:
A = 10 [Number of eggs in the first basket]
B = 2 [Number of eggs in the second basket]
T = Total number of eggs

Equations:
T = A + B. The total number of eggs is the sum of the eggs in the first and second baskets.

Goal: T. How many eggs are there total?

The above example has a wrong variable value and missed an equation. "There are twice as many eggs in the second basket as the first" should be translated into B = 2 * A instead of B = 2.

INCORRECT EXAMPLE 3

Variables:
T [Total number of eggs]

Equations:
T = 10 + 20. The total number of eggs is the sum of the 10 eggs in the first basket and the 20 eggs in the second basket.

Goal: T. How many eggs are there total?

The above example combines too many operations into a single equation, in a way that is not faithful to the original question. A good rule of thumb is to have one variable stand in for every number in the problem, and have all equations be of one of two forms: (1) assigns one variable to one constant, or (2) assigns one variable to a relation among other variables. Avoid equations that can be simplified.

**Math Problem**

A notebook costs twice as much as a pencil and an eraser combined. A pen costs as much as a pencil and eraser combined. A pencil costs $1.20 and an eraser costs $0.30. How much will 8 pens cost?

**Evaluation Criteria**

- Try to solve this problem. What is the answer?

| 12 | Proceed To Translation |

☐ Not sure

Please translate the above math problem into a list of variables, a list of equations, and a goal variable.

**List of variables in the above problem**

| Variable name (e.g. A) | Variable value (e.g. 10, or leave it empty if the value is not directly mentioned or common knowledge, and must be computed from the other variables) | Variable definition (e.g. "number of eggs in the first basket") | Is the variable irrelevant to the final answer? Check off the checkbox next to the variable if it is irrelevant. (E.g. "R" should be checked off and nothing else.) | + |
|---|---|---|---|---|
| | + − | | ☐ | X |

**List of equations in the above problem**

All equations must be a combination of the following and **no other symbols or characters**

1. The variables defined above
2. Integers and decimals
3. Operations, as standardized below:
   * for times (multiply)
   + for plus
   - for minus
   / for divides
   = for equals
   () for parentheses
   You may use more operations than the ones defined above if needed, but if the operation you need is defined above, please use the appropriate symbol.
4. Do NOT include any units ($, hours, days, etc.)
5. Remember to include * any time you are performing multiplications, e.g. use 2 * A and not 2A.
6. Convert all percentages into decimals, e.g. 150% becomes 1.5

| Equation (e.g. T = A + B) | Equation verbal description (e.g. "The total number of eggs is the sum of the number of eggs in the first basket and the number of eggs in the second basket.") | + |
|---|---|---|
| | | X |

**Goal variable:**

| Goal variable (e.g. T) | Goal description (e.g. "how many eggs are there total?") |
|---|---|
| C2 | Cost of 8 pens |

Figure 4: Screenshot of the annotation interface used for obtaining CSPs for each math problem in the GSM setting.

## C.4 GSM-Q

We ask annotators to translate problems from GSME-Q into word problems. Full instructions we provided to annotators can be found below with interface screenshot in Figure 5.

You will be presented with a series of math problems. These math problems are written in words and translated to equations. Your task is to first validate whether the translation is correct given the information present in the problem. If so, you will then be prompted to answer questions for each equation. An example is provided below. Please read carefully and make sure you understand before proceeding.

Math problem written in words:
*If there are 10 eggs and 5 ribbons in a basket, and there are twice as many eggs in the second basket as the first basket, how many eggs are there total?*

The following is the math problem translated into equations, together with the variables that appear in the equations and the goal of the problem.

Variables:
E1 = 10 [Number of eggs in the first basket]
E2 [Number of eggs in the second basket]
R = 5 [Number of ribbons in the first basket]
T [Total number of eggs]

Equations:
E2 = 2 * E1 [There are twice as many eggs in the second basket as the first.]
T = E1 + E2 [The total number of eggs is the sum of the eggs in the first and second baskets.]

Goal:
T [How many eggs are there total?]

You will first validate that the variables, equations, and goal matches the original problem described in words:

1. Is the above list of variables, equations, and the goal equivalent to the original math problem written in words?:
    (a) [Bubble] Yes
    (b) [Bubble] No
In the example above, the answer is "yes". Here are some examples of wrong translations, where the answer would be "no":
INCORRECT EXAMPLE 1

    Variables:
    E1 = 10 [Number of eggs in the first basket]
    T [Total number of eggs]

    Equations:
    T = E1 + E2. The total number of eggs is the sum of the eggs in the first and second baskets.

    Goal:
    T [How many eggs are there total?]
The above example is missing a variable (the number of eggs in the second basket) that is necessary to solve for the goal
    INCORRECT EXAMPLE 2
    Variables:
    E1 = 10 [Number of eggs in the first basket]
    E2 = 2 [Number of eggs in the first basket]
    T [Total number of eggs]

    Equations:
    T = E1 + E2 [The total number of eggs is the sum of the eggs in the first and second baskets.]

Goal:
T [How many eggs are there total?]

The above example has a wrong variable assignment. "There are twice as many eggs in the second basket as the first" should be translated into E2 = 2 * E1 instead of E2 = 2.

INCORRECT EXAMPLE 3

Variables:
T [Total number of eggs]

Equations:
T = 10 + 20 [The total number of eggs is the sum of the 10 eggs in the first basket and the 20 eggs in the second basket.]

Goal:
T [How many eggs are there total?]

The above example combines too many operations into a single equation, in a way that is not faithful to the original question.
A good rule of thumb is to have one variable stand in for every number in the problem, and have all equations be of one of two forms: (1) assigns one variable to one constant, or (2) assigns one variable to a relation among other variables. Avoid equations that can be simplified.

If you answered "yes" to the above question, you will see the following questions.

2. Please solve for the "Goal" in the above list of variables and equations. Is your answer the same as the answer below?:
30
    (a)  [Bubble] Yes
    (b)  [Bubble] No
In the example above, the answer is yes. In incorrect example 1, the answer would be "no" (the goal value is ambiguous in incorrect example 1). In incorrect example 2, the answer would also be "no" (the goal value is 12 in incorrect example 2).

If you answered "no" to the above question, the questions will end here. If you answered "yes", then, you will answer the following questions for different sets of equation(s):

3. Try to rewrite the problem to remove all parts of the problem that states any of the above equation(s). Please make sure the problem is still coherent English (e.g., do not simply delete the section you copied above without fixing any grammatical errors). Please also make sure to remove the entire premise, not just replacing numbers with "few" or "some". If there is no way to remove the equation (e.g., because it was not mentioned in the original problem), please leave the text box empty and check off "cannot remove".

[Text area]

    ☐ Cannot remove
For example, given:
    • E1 = 10 [Number of eggs in the first basket]
    • R = 5 [Number of ribbons in the first basket]
you may write (a) "If there are twice as many eggs in the second basket as the first basket, how many eggs are there total?"
Given:
    • E2 = 2 * E1 [There are twice as many eggs in the second basket as the first.]
    • R = 5 [Number of ribbons in the first basket]
you may write (b) "If there are 10 eggs in a basket, how many eggs are there total?"
Given:
    • T = E1 + E2 [The total number of eggs is the sum of the eggs in the first and second baskets.]
    • R = 5 [Number of ribbons in the first basket]
you would check off "Cannot Remove" because "T = E1 + E2" was never mentioned in words in the prompt, but was implied.

4. Given the above rewritten problem, is the answer to the question:

**Math Problem**

There are 220 castles in Scotland. 40 percent of them are manned, 40 percent of them are ruined, and half of the ruined castles are unmanned. How many unmanned ruined castles are there in Scotland?

**Below, we translate this problem into variables, equations and the goal.**

Variables:
A = 220 [Total number of castles in Scotland]
B [Number of ruined castles]
D [Number of unmanned ruined castles]
E = 0.4 [Percentage of the total castles that are ruined]
F = 0.5 [Percentage of the ruined castles that are unmanned]
G = 0.4 [Percentage of the total castles that are manned]

Equations:
B=E*A [Number of ruined castles is 40% of the total castles]
D=F*B [Half of the ruined castles are unmanned]

Goal:
D [Number of unmanned ruined castles]

**Evaluation Criteria**

- Is the above list of variables, equations, and the goal equivalent to the original math problem written in words?
  - ◉ Yes
  - ○ No
- Please solve for the "Goal" in the above list of variables and equations. Is your answer the same as **44** ?
  - ◉ Yes
  - ○ No

A = 220 [Total number of castles in Scotland] G = 0.4 [Percentage of the total castles that are manned]

- Try to rewrite the problem to **remove all** parts of the problem that states any of the above equation(s). Please make sure the problem is still coherent English (e.g. do not simply delete the section you copied above without fixing any grammatical errors). Please also make sure to **remove the entire premise**, not just replacing numbers with "few" or "some". If there is no way to remove the equation (e.g. because it wasn't mentioned in the original problem), please leave the text box empty and check off "cannot remove".

  > There are castles in Scotland. percent of them are manned, 40 percent of them are ruined, and half of the ruined castles are unmanned. How many unmanned ruined castles are there in Scotland?

  ☐ Cannot remove

- Given the above rewritten problem, is the answer to the question:
  - ○ still **44**
  - ◉ Unclear
  - ○ Different from **44**

Figure 5: Screenshot of the annotation interface used for obtaining underspecified GSM problems written in words.

☐ the same as your answer in question 2
☐ unclear
☐ different from your answer in question 2. What is the new answer?: [Text box]
In the example above, the answer would likely be "unclear" for question (a). For question (b), the answer would likely be "different", with the new answer being 10.

# D Prompts from each domain in QUESTBENCH

## D.1 Identifying missing information and asking the right question

**Logic-Q.** We used the following prompt for Logic-Q:

> Suppose you know the following rules about Alice:
> {rules_nl}
>
> You trying to discern whether a statement about Alice is true given some facts. You must decide whether you have enough information to determine whether the final statement is true. You may respond with one of the following-
> If you do not have enough information yet, you may ask a question about an attribute of Alice, in the form of "Question: Is Alice [attribute]?". Ask the best question that, regardless of how it is answered, provides the most information about the final statement.
> Once you have enough all information necessary to determine the truth value of the statement, you can terminate with "End questioning".
> Generate one of "Question: Is Alice [attribute]?" or "End questioning" and nothing else.

**Planning-Q.** We used the following prompt for Planning-Q:

> You will be given a planning problem in the domain defined by the following PDDL:
>
> {domain_pddl}
>
> The current objects are present in the problem:
>
> {problem_objects}
>
> You will be given a set of conditions true in your initial state and a set of goal conditions, and will need to construct a plan from your current state to the goal state. Some details of your initial state may be missing. You must decide whether you have enough information to disambiguate a plan to the final state. If not, you must decide what information is necessary to construct a fully unambiguous plan from your initial state to the goal state. You will be presented with a set of multiple-choice options for questions you may ask, and you must answer with one of the options. Please reason step-by-step, then generate "Answer:" followed by the number of the option and nothing else.
>
> Known facts about current state:
> {conditions}
>
> Goal state:
> {goals}
>
> Possible questions:
> 0. Is {proposition_0} true?
> 1. Is {proposition_1} true?
> 2. Is {proposition_2} true?
> ...

**GSM-Q.** We used the following prompt for GSM-Q:

> You are trying to solve a math problem. You must decide whether you have enough information to solve the math problem. Please respond with one of the following-
> If you do not have enough information to solve the math problem, you may ask a question back to the user from a set of predefined "Possible questions". Otherwise, choose "No questions needed."
> Generate the number of your choice in the form "Choice: number"
>
> Math problem: {problem}
>
> Possible questions:
> 0. What is {variable_name} ({variable_symbol})?
> 1. What is {variable_name} ({variable_symbol})?

```
...
{n}. No questions needed.
```

**GSME-Q.**   We used the following prompt for GSME-Q:

```
You are trying to solve a math problem. You must decide whether you have enough information to
solve the math problem. Please respond with one of the following-
If you do not have enough information to solve the math problem, you may ask a question back to
the user from a set of predefined "Possible questions". Otherwise, choose "No questions needed."
Generate the number of your choice in the form "Choice: number"

Math problem: {problem }
Possible questions:
0. What is the value of {variable_symbol}?
1. What is the value of {variable_symbol}?
{n}. No questions needed.
```

## D.2   Reasoning in the presence of sufficient information

**Logic-Q.**   We used the following prompt for Logic-Q:

```
Suppose you know the following rules about Alice:
{rules_nl}

You will presented with a binary question about an attribute of Alice. Please answer it with "Yes" or
"No".
```

**Planning-Q.**   We used the following prompt for Planning-Q:

```
You will be given a planning problem in the domain defined by the following PDDL:

{domain_pddl}

The current objects are present in the problem:

{problem_objects}

You will be given a set of conditions true in your initial state and a set of goal conditions, and will
need to construct a plan from your current state to the goal state.
Please answer with "Answer:" followed by the plan in the form of a comma-separated list of ([action]
[args]) (e.g. (pick-up a), (stack a b), ...).

Known facts about current state:
{conditions}

Goal state:
{goals}
```

**GSME-Q/GSM-Q.**   We used the following prompt for GSME-Q/GSM-Q:

```
You are trying to answer a math question. Please answer with "Answer:" followed by the answer to
the math question. Only include the raw numerical answer, do not include any units or thousands
separators.

Math problem: {problem}
```

## D.3   Detecting underspecification and recognizing uncertainty

**Logic-Q.**   We used the following prompt for Logic-Q:

|  |  | Logic-Q | Planning-Q | GSME-Q | GSM-Q |
|---|---|---|---|---|---|
| ZS | Gemini Flash Thinking 2.0 Exp 01-21 | 89.34% | 53.46% | 96.03% | 98.26% |
|  | Claude 3.5 Sonnet | 64.08% | 51.26% | 29.14% | 60.76% |
|  | GPT-4o | 51.93% | 24.53% | 26.49% | 38.54% |
|  | Gemini 1.5 Pro | 70.19% | 21.33% | 35.10% | 17.36% |
|  | Gemini 1.5 Flash | 71.34% | 12.64% | 17.22% | 6.25% |
|  | Gemma 2 2B IT | 50.76% | 0.01% | 4.64% | 8.33% |
|  | Gemma 2 9B IT | 51.06% | 0.94% | 9.93% | 18.06% |
|  | Gemma 2 27B IT | 56.18% | 1.25% | 16.56% | 2.43% |
| ZS + CoT | Gemini Flash Thinking 2.0 Exp 01-21 | 88.64% | 53.45% | 96.03% | 98.26% |
|  | Claude 3.5 Sonnet | 78.59% | 69.54% | 100.00% | 100.00% |
|  | GPT-4o | 76.33% | 14.12% | 100.00% | 96.18% |
|  | Gemini 1.5 Pro | 74.74% | 23.48% | 96.03% | 95.14% |
|  | Gemini 1.5 Flash | 75.21% | 12.84% | 100.00% | 92.36% |
|  | Gemma 2 2B IT | 56.30% | 0.00% | 78.15% | 54.51% |
|  | Gemma 2 9B IT | 58.75% | 0.62% | 75.50% | 77.78% |
|  | Gemma 2 27B IT | 53.97% | 0.49% | 88.74% | 77.78% |
| 4S | Gemini Flash Thinking 2.0 Exp 01-21 | 81.23% | 48.29% | 100.00% | 97.92% |
|  | Claude 3.5 Sonnet | 65.80% | 54.54% | 58.28% | 70.83% |
|  | GPT-4o | 58.99% | 30.61% | 39.07% | 65.28% |
|  | Gemini 1.5 Pro | 55.41% | 32.74% | 40.40% | 45.83% |
|  | Gemini 1.5 Flash | 57.52% | 38.32% | 22.52% | 47.57% |
|  | Gemma 2 2B IT | 51.88% | 0.05% | 13.25% | 43.40% |
|  | Gemma 2 9B IT | 62.06% | 6.11% | 22.52% | 54.51% |
|  | Gemma 2 27B IT | 56.05% | 4.93% | 29.14% | 65.97% |

Table 6: Language model accuracies for answering questions correctly in a well-specified variant of QUESTBENCH. See Section 4.3 for more information.

---

Suppose you know the following rules about Alice: {rules_nl}
You will presented with a binary question about an attribute of Alice. Please answer it with "Yes" or "No" or "Not sure".

---

**Planning-Q.** We used the following prompt for Planning-Q:

---

You will be given a planning problem in the domain defined by the following PDDL:
{domain_pddl}
The current objects are present in the problem:
{problem_objects}
You will be given a set of conditions true in your initial state and a set of goal conditions, and will need to construct a plan from your current state to the goal state. Please answer with "Answer:" followed by the plan in the form of a comma-separated list of ([action] [args]) (e.g. (pick-up a), (stack a b), ...), or "Not sure" if you are unsure what the plan should be.

---

**GSME-Q/GSM-Q.** We used the following prompt for GSME-Q/GSM-Q:

---

You are trying to answer a math question. Please answer with "Answer:" followed by the answer to the math question, or "Not sure" if you are not sure what the answer is. Only include the raw numerical answer, do not include any units or thousands separators.
Math problem: {problem}

---

# E   Search solutions for Logic-Q and Planning-Q

## E.1   Logic-Q

**Brute-force Solution.**   A problem in Logic-Q can be solved through brute force search as follows. We have a subroutine `infer` that allows us to infer the values of all variables consistent with the current variable assignments. For example, if $A = \{a, b\}$ and we know that $a \wedge b \rightarrow c$, then $\mathtt{infer}(A)$ gives us $\{a, b, c\}$. At a high-level, `infer` is analogous to breadth-first-search, and thus has runtime complexity $O(|X| + |C|)$.

To solve a Logic-Q problem, we first run `infer` to get values of all variables consistent with the current assignment $A$, creating $A'$. Next, for all unassigned variables $x_u \in X \backslash A'$, we iterate through them one at a time and check whether:

1. Including $x_u$ in $A'$ allows us to infer the target variable is either *true* or *false*.
2. Include $\neg x_u$ in $A'$ allows us to infer the target variable is the *opposite assignment* as it was in case (1).

The first time 1 and 2 both hold, we have that $x_u$ is true.

The expected runtime of brute-force can thus be computed by the expected number of variables that we need to iterate through to get to a variable in the sufficient set (upperbounded by $\mathbb{E}_{\mathrm{BF}}$ in §2.4), multiplied by the complexity of running the `infer` algorithm twice. By Markov's inequality, with probability $\geq 1 - \delta$, the complexity is bounded by

$$O((|X| + |C|)\mathbb{E}_{\mathrm{BF}}/\delta).$$

**Backwards Search Solution.**   A problem in Logic-Q can be solved through backwards search. The procedure is the same as the backwards search used to construct the dataset, described in Section C.1.1. At each iteration, we keep track of a disjunction of conjunction of variables required to prove $y$.

The complexity of backwards search is given by the search depth $d$ multiplied by the number of expansions per depth, which is bounded by $O(3^{|X|}|C|)$ (expanding up to $|C|$ rules for conjunction, for which there are at most $3^{|X|}$ conjunctions – each variable can be either true/false/missing from that conjunction) Thus, the overall complexity is $O(3^{|X|}|C|d)$.

Though comprehensive backwards search is EXP-time, we can terminate early as soon as we find a disjunction consisting of the negation of all initial conditions, the goal variable, and an additional term, which would mean that asking about the value of the initial term is sufficient to infer the value of the goal variable.

If we know this disjunction is at most at depth $d$, then the runtime is bounded by $|X|^{|C|d}$, where $|X|^{|C|}$ is the branching factor at each node. The branching factor comes from the cross product of at most $|X|$ terms across at most $|C|$ conjunctive constraints. Thus, the total runtime is given by

$$O(|X|^{|C|d}).$$

## E.2   Planning-Q

**Brute-force Solution.**   As described in §2.4, our brute-force approach randomly samples from the choices without replacement, and checks each sampled choice until a correct one is found. The complexity of this approach relies on the complexity of validating whether a choice is correct or not.

Given an underspecified initial state $\tilde{s}_0$ and a selected choice of atom $x$, we can generate all physically-plausible initial states $s_0^+ \in F(\tilde{s}_0, x)$ and $s_0^- \in F(\tilde{s}_0, \neg x)$, where function $F$ maps from a partial state to all states consistent with the partial state. We can then solve the planning problem for *all* initial states in both sets, and check whether the following holds.

1. The shortest path $\hat{\tau}^+$ remains the same for every initial $s_0^+ \in F(\tilde{s}_0, x)$.
2. The shortest path $\hat{\tau}^-$ remains the same for every initial $s_0^- \in F(\tilde{s}_0, \neg x)$.
3. $\hat{\tau}^+$ and $\hat{\tau}^-$ are different.

Assume the planning tool we use is breadth-first search, which takes at most $O(|S| + |S||\mathcal{A}|)$ time. We perform breadth-first search at most $|S|$ times (since $|F(\tilde{s}_0, \neg x) \cup F(\tilde{s}_0, \neg x)| < |S|$), meaning

the overall runtime of this solution is $O((|S| + |S||\mathcal{A}|)|S|)$. The number of states is bounded by $|S| = 2^{|X|}$, while the number of actions is bounded by $|\mathcal{A}| = 2b + 2b(b+1) = O(b^2)$ where $b$ is the number of blocks in the domain, so the overall complexity of validating a choice is $O(2^{2|X|}b^2)$.

Similar to §E.1, with probability at least $1 - \delta$, we only need to guess $\mathbb{E}_{BF}/\delta$ times. So the overall complexity is

$$O(2^{2|X|}b^2 \mathbb{E}_{BF}/\delta).$$

**Backward Search Solution.** We replace the breadth-first-search from each consistent state with a single backwards search. Starting from the goal condition, we iterate backwards to find sets of initial states that utilize the same path to the goal. This is the same backwards search that was used to construct all well-specified partial states, described in Section C.2.1. We expand backwards until we arrive at a set of partial initial states that are consistent with the given partially-observed set $\tilde{s}_0$, and we find the attribute that distinguishes each partial initial state from each other.

Because the number of partial states is bounded by $3^{|X|}$ (each proposition can take on 3 values: true/false/unknown), the backwards breadth-first search takes $O(3^{|X|} + 3^{|X|}|\mathcal{A}|)$. However, if we know that the longest path from any initial state consistent with $\tilde{s}_0$ to the goal is of length $d$, meaning we only need to search up to depth $d$, then the runtime is bounded by

$$O(|\mathcal{A}|^d) = O(b^{2d}) \text{ or } O(3^{|X|d})$$

(Each iteration, expand at most $3^{|X|}$ states or $|\mathcal{A}|$ actions.)

# F  Qualitative Study of QuestBench Reasoning Traces

While the Gemini 2.0 Flash Thinking Experimental model that we used during the original set of evaluations is no longer available at the time we conduct these analyses, we include some reasoning traces from Gemini 2.5 Pro (Google's newest reasoning model as of October 2025) on a Logic-Q and Planning-Q problem, respectively.

## F.1  Logic-Q

We use the first example in the Logic-Q subset of QuestBench, as follows:

[SYSTEM]
Suppose you know the following rules about Alice:

If Alice is aggressive and difficult and rational, then Alice is supportive.
If Alice is aggressive and hilarious and light, then Alice is unpleasant.
If Alice is aggressive and hurt, then Alice is calm.
If Alice is aggressive and loving, then Alice is versatile.
If Alice is aggressive, then Alice is difficult.
If Alice is aggressive, then Alice is elated.
If Alice is amused and dull and sensible, then Alice is hurt.
If Alice is amused and hypocritical, then Alice is elated.
If Alice is amused and precious, then Alice is difficult.
If Alice is amused and sleepy, then Alice is rational.
If Alice is calm and grumpy and light, then Alice is frightened.
If Alice is calm and loving, then Alice is dull.
If Alice is calm, then Alice is silly.
If Alice is different and fearless and versatile, then Alice is elated.
If Alice is different and hypocritical, then Alice is unpleasant.
If Alice is different and powerful, then Alice is messy.
If Alice is different and sensible and shiny, then Alice is rational.
If Alice is different and shiny, then Alice is tired.
If Alice is different and thoughtless, then Alice is sensible.
If Alice is different, then Alice is sensible.
If Alice is difficult and hypocritical, then Alice is versatile.
If Alice is difficult and loving and versatile, then Alice is thoughtless.
If Alice is difficult and precious, then Alice is tired.
If Alice is difficult and shiny, then Alice is elated.
If Alice is difficult and sleepy, then Alice is dull.
If Alice is dull and frightened and sensible, then Alice is elated.
If Alice is dull and light, then Alice is different.
If Alice is dull and loving and messy, then Alice is powerful.
If Alice is dull and precious and rational, then Alice is modern.
If Alice is elated and rational and tired, then Alice is sensible.
If Alice is elated and supportive and thoughtless, then Alice is aggressive.
If Alice is elated, then Alice is messy.
If Alice is fearless and grumpy and shiny, then Alice is frightened.
If Alice is fearless and messy, then Alice is frightened.
If Alice is fearless, then Alice is amused.
If Alice is fearless, then Alice is shiny.
If Alice is frightened and hurt and precious, then Alice is supportive.
If Alice is frightened and rational, then Alice is different.
If Alice is frightened and silly, then Alice is calm.
If Alice is frightened, then Alice is unpleasant.
If Alice is grumpy and hurt and powerful, then Alice is elated.
If Alice is grumpy, then Alice is shiny.
If Alice is hilarious and hurt and modern, then Alice is shiny.
If Alice is hilarious and unpleasant, then Alice is fearless.
If Alice is hilarious, then Alice is hurt.
If Alice is hurt and hypocritical, then Alice is silly.
If Alice is hurt and light and unpleasant, then Alice is amused.
If Alice is hurt and light and unpleasant, then Alice is supportive.

If Alice is hurt and messy and modern, then Alice is calm.
If Alice is hurt and messy and supportive, then Alice is sleepy.
If Alice is hurt and sensible and unpleasant, then Alice is shiny.
If Alice is hurt and silly, then Alice is frightened.
If Alice is hypocritical and shiny and thoughtless, then Alice is dull.
If Alice is loving, then Alice is powerful.
If Alice is loving, then Alice is silly.
If Alice is modern, then Alice is loving.
If Alice is rational and shiny, then Alice is modern.
If Alice is sensible, then Alice is rational.
If Alice is shiny and silly and unpleasant, then Alice is hurt.
If Alice is shiny, then Alice is powerful.
If Alice is sleepy and versatile, then Alice is calm.
If Alice is sleepy and versatile, then Alice is sensible.
If Alice is supportive and tired, then Alice is elated.
If Alice is tired, then Alice is sensible.
If Alice is unpleasant, then Alice is modern.

You trying to discern whether a statement about Alice is true given some facts. You must
    decide whether you have enough information to determine whether the final statement is
    true. You may respond with one of the following– \\
If you do not have enough information yet, you may ask a question about an attribute of Alice,
    in the form of "Question: Is Alice [attribute]?". Ask the best question that, regardless of
    how it is answered, provides the most information about the final statement.
Once you have enough all information necessary to determine the truth value of the statement
    , you can terminate with "End questioning". \\
Generate one of "Question: Is Alice [attribute]?" or "End questioning" and nothing else.

[USER]
Alice is amused.
Alice is calm.
Alice is loving.
Alice is sleepy.

You may not ask if Alice is sensible.
Is Alice tired?

We put the above prompt into Gemini 2.5 Pro, and receive the following (summarized) reasoning
trace:

Assessing the Task

I'm currently focused on the core objective: acting as a logic engine to determine the truth of
    specific statements about Alice, based on given rules and facts. I've broken down the
    process into analyzing the goal, understanding the rules, and applying them. I'm
    prioritizing accuracy and efficiency in this logical deduction.

Unfolding Deductions

I'm now iterating through the rules, attempting to deduce "Alice is tired". I've mapped the initial
    facts and am applying rules like "If Alice is amused and dull and sensible, then Alice is
    not tired". My goal is to use forward chaining from the initial facts. So far, the process
    indicates I need to fill in some gaps in knowledge; hence the questions. I continue to
    check if "Alice is tired" or its logical complement can be derived.

Elaborating Connections

I'm expanding my fact–checking process. I've incorporated the new facts "rational", "dull", "silly
    ", and "powerful". I'm now cross–referencing these against the rules to check for new
    implications. I'm keenly aware of the constraint preventing me from asking about "

sensible", so I'm carefully navigating around any rules involving that attribute. My goal remains the same: to determine if "Alice is tired" is provable or not.

### Evaluating Derived Truths

I'm now conducting a second round of rule application, using the facts I derived previously – rational, dull, silly, and powerful. I'm focusing on rules where these facts could lead to a conclusion about whether Alice is tired. The constraint about "sensible" continues to guide my search, ensuring I don't follow any paths that require this attribute. My primary task remains to determine if "Alice is tired" can be proven or disproven with the available information.

### Analyzing Implications

I'm now methodically applying the rules to the established facts, looking for further inferences. I've successfully derived "rational", "dull", "silly", and "powerful" in the first pass. I'm focusing on "tired" and the attributes that could lead to it. The "sensible" constraint is a constant consideration, steering my focus away from any rules dependent on it. The task is to exhaust all possible deductions within the given constraints.

### Refining the Deduction

I'm now employing backward chaining, examining the rules to see how "tired" can be derived. I 've considered rules like "If different and shiny, then tired" and "If difficult and precious, then tired." However, I need to evaluate if I can prove these. Since I lack the ability to inquire about "sensible", I can't derive some relevant facts. My task persists: to determine if "Alice is tired" is provable.

### Tracing Deductions

I've explored backward chaining to analyze how "tired" can be derived. Examining the rules, I focused on implications like "If different and shiny, then tired" and "If difficult and precious , then tired". I've initiated a path to find the truth values of those conditions. The "sensible " constraint is a constant consideration. I'm actively working through each path to reach a conclusion regarding "Alice is tired".

### Reflecting on Current Path

I'm now fully immersed in the deduction process, focusing on determining whether "Alice is tired" is provable. I've considered all rules, both forward and backward, and the initial facts. I've systematically applied the rules and, due to the constraint regarding "sensible", have had to adapt my strategies. My goal is to exhaustively evaluate every possible inference path to achieve a decisive answer.

### Evaluating Next Steps

I've explored forward and backward chaining, identifying key paths to deduce "Alice is tired." I' m now formulating a prioritized question strategy, keeping the "sensible" constraint in mind. My current aim is to minimize the number of questions needed to reach a definitive conclusion.

### Advancing Deduction Strategy

I'm adapting my questioning approach to focus on the most direct paths to proving "Alice is tired." I'm prioritizing asking if Alice is "precious" as a key first step based on rule analysis. The constraint about "sensible" continues to guide my reasoning and the ordering of possible questions.

### Optimizing Question Strategy

I'm focusing my approach on the most efficient questioning path. I've narrowed it down to prioritizing queries that can directly influence the conclusion "Alice is tired." My strategy now hinges on finding the most informative question given the rules and constraints.

The model answers incorrectly: the correct options are asking about Alice being "different", "frightened", or "hurt". Based on the reasoning trace, the model begins by pursuing forward search ("Unfolding Deductions", "Evaluating Derived Truths", and "Analyzing Implications"). Next, it switches to backwards breadth-first search ("Refining the Deduction", and "Tracing Deductions"). It claims to have exhausted all the rules in "Reflecting on Current Path" (though we lack the full reasoning trace to verify this), and identified all the possible inference paths to the target attribute. It then chooses its final question based on "quickest path to a conclusion". It appears that while the LM thoroughly investigates the positive direction (what is needed to establish that Alice *is* tired), it never investigates the negative direction (what is needed to establish that Alice is *not* tired). Thus, it did not identify the most informative question: unlike the ground truth questions, which would've been sufficient to establish the target attribute regardless of whether they were answered positively or negatively, precious is only sufficient to establish the target attribute being true if precious were true, but is insufficient to establish the target attribute being false if precious were false.

## F.2 Planning-Q

We use the following example in the Planning-Q subset of QuestBench:

```
[SYSTEM]
You will be given a planning problem in the domain defined by the following PDDL:

;;;;;;;;;;;;;;;;;;;;;;;;;;;;;;;;;;;;
;;;;;;;;;;;;;;;;;;;;;;;;;;;;;;;;;;;;
;;; 4 Op–blocks world
;;;;;;;;;;;;;;;;;;;;;;;;;;;;;;;;;;;;
;;;;;;;;;;;;;;;;;;;;;;;;;;;;;;;;;;;;

(define (domain BLOCKS)
  (:requirements :strips :typing)
  (:types block)
  (:predicates (on ?x – block ?y – block)
         (ontable ?x – block)
         (clear ?x – block)
         (handempty)
         (holding ?x – block)
         )

  (:action pick–up
```

```
            :parameters (?x – block)
            :precondition (and (clear ?x) (ontable ?x) (handempty))
            :effect
            (and (not (ontable ?x))
                (not (clear ?x))
                (not (handempty))
                (holding ?x)))

  (:action put–down
            :parameters (?x – block)
            :precondition (holding ?x)
            :effect
            (and (not (holding ?x))
                (clear ?x)
                (handempty)
                (ontable ?x)))
  (:action stack
            :parameters (?x – block ?y – block)
            :precondition (and (holding ?x) (clear ?y))
            :effect
            (and (not (holding ?x))
                (not (clear ?y))
                (clear ?x)
                (handempty)
                (on ?x ?y)))
  (:action unstack
            :parameters (?x – block ?y – block)
            :precondition (and (on ?x ?y) (clear ?x) (handempty))
            :effect
            (and (holding ?x)
                (clear ?y)
                (not (clear ?x))
                (not (handempty))
                (not (on ?x ?y)))))
```

The current objects are present in the problem:

['a', 'b', 'c', 'd', 'e']

You will be given a set of conditions true in your initial state and a set of goal conditions, and will need to construct a plan from your current state to the goal state.
Some details of your initial state may be missing. You must decide whether you have enough information to disambiguate a plan to the final state. If not, you must decide what information is necessary to construct a fully unambiguous plan from your initial state to the goal state.
You will be presented with a set of multiple–choice options for questions you may ask, and you must answer with one of the options.
Please generate the number of the option and nothing else.

[USER]
Known facts about current state:
(clear a)
(handempty)
(on a e)
(on b d)
(on e b)
(ontable c)

Goal state:
(on b a)
(on c b)
(ontable a)

Possible questions:
0. Is (clear a) true?
1. Is (clear b) true?
2. Is (clear c) true?
3. Is (clear d) true?
4. Is (clear e) true?
5. Is (handempty) true?
6. Is (holding a) true?
7. Is (holding b) true?
8. Is (holding c) true?
9. Is (holding d) true?
10. Is (holding e) true?
11. Is (on a b) true?
12. Is (on a c) true?
13. Is (on a d) true?
14. Is (on a e) true?
15. Is (on b a) true?
16. Is (on b c) true?
17. Is (on b d) true?
18. Is (on b e) true?
19. Is (on c a) true?
20. Is (on c b) true?
21. Is (on c d) true?
22. Is (on c e) true?
23. Is (on d a) true?
24. Is (on d b) true?
25. Is (on d c) true?
26. Is (on d e) true?
27. Is (on e a) true?
28. Is (on e b) true?
29. Is (on e c) true?
30. Is (on e d) true?
31. Is (ontable a) true?
32. Is (ontable b) true?
33. Is (ontable c) true?
34. Is (ontable d) true?
35. Is (ontable e) true?
36. No questions needed.

The Gemini 2.5 Pro reasoning trace on this problem can be found below:

Understanding the Objective

I've been examining the target state – a neat tower of blocks. It's clear the solution requires arranging c atop b, which sits on a, that last block positioned on the table. The essence of the problem is now defined.

Mapping the States

I'm now mapping the initial and goal states. The initial state is a single, unstable tower: a–e–b–d, with c on the table. 'a' is clear, and we have an empty hand. The goal state, the neat tower, requires 'c' on 'b', 'b' on 'a', and 'a' on the table. The problem is clear. I see I must dismantle the existing towers, and reassemble them.

Disambiguating the Variables

I'm now focusing on the uncertainties, mainly whether d is on the table or if c has something on top of it. I've laid out the possible scenarios, and I'm realizing that the true initial state is actually ambiguous. The plan must work regardless of whether there's an additional block on c or if d is on the table.

Resolving Ambiguities

I've been working through the uncertainties in the initial state, especially the positions of
blocks 'c' and 'd'. The analysis reveals that if 'd' is on the table, 'c' must be clear, and vice
versa. I'm now structuring the plan to handle either scenario effectively.

Structuring the Solution

The initial and goal states are now clarified, and I've created a mental model. I recognize the
essential tasks are to dismantle the initial tower and reassemble the blocks to meet the
goal condition, c on b on a. I've outlined the necessary moves using unstack, put–down,
and stack commands, considering the constraints of an empty hand. The challenge is in
determining the initial position of block 'd', which is linked to whether 'c' is clear or not.

Clarifying the Initial State

Now I'm diving deeper into the initial conditions, pinpointing the two key uncertainties: the
location of 'd' and the clarity of 'c'. Analyzing both scenarios confirms that resolving either
uncertainty automatically resolves the other, simplifying the planning process.

Planning the Execution

I've streamlined the plan based on the clarified initial state analysis. The strategy focuses on
dismantling the tower, placing each block on the table, then reassembling the blocks to
fulfill the goal. I'm now testing the actions required for each scenario, and I'm ready to
move forward.

Defining the Initial States

I'm now zeroing in on the initial state's subtleties. It's become clear that the position of 'd', and
consequently the clarity of 'c', is the crux. I'm building a concise plan that seamlessly
addresses both possible scenarios – 'd' on the table versus 'd' atop 'c'. I've identified the
key uncertainties that inform the planning process.

2

In this case, the model correctly identifies a minimal piece of resolving information: Is (clear c) true?
The model appears to be using a non-search-based approach of visualizing the state of the table and
the stacks of blocks, which may explain why LLM performance only weakly correlated with search
complexity factors in this domain (see Section 4.2).

# G    Graphs of Correlation between Problem Complexity and Clarification Question Accuracy

Full graphs of the correlation between LLM performance on QUESTBENCH and the factors listed in Section 2.4 can be found in Figure 6.

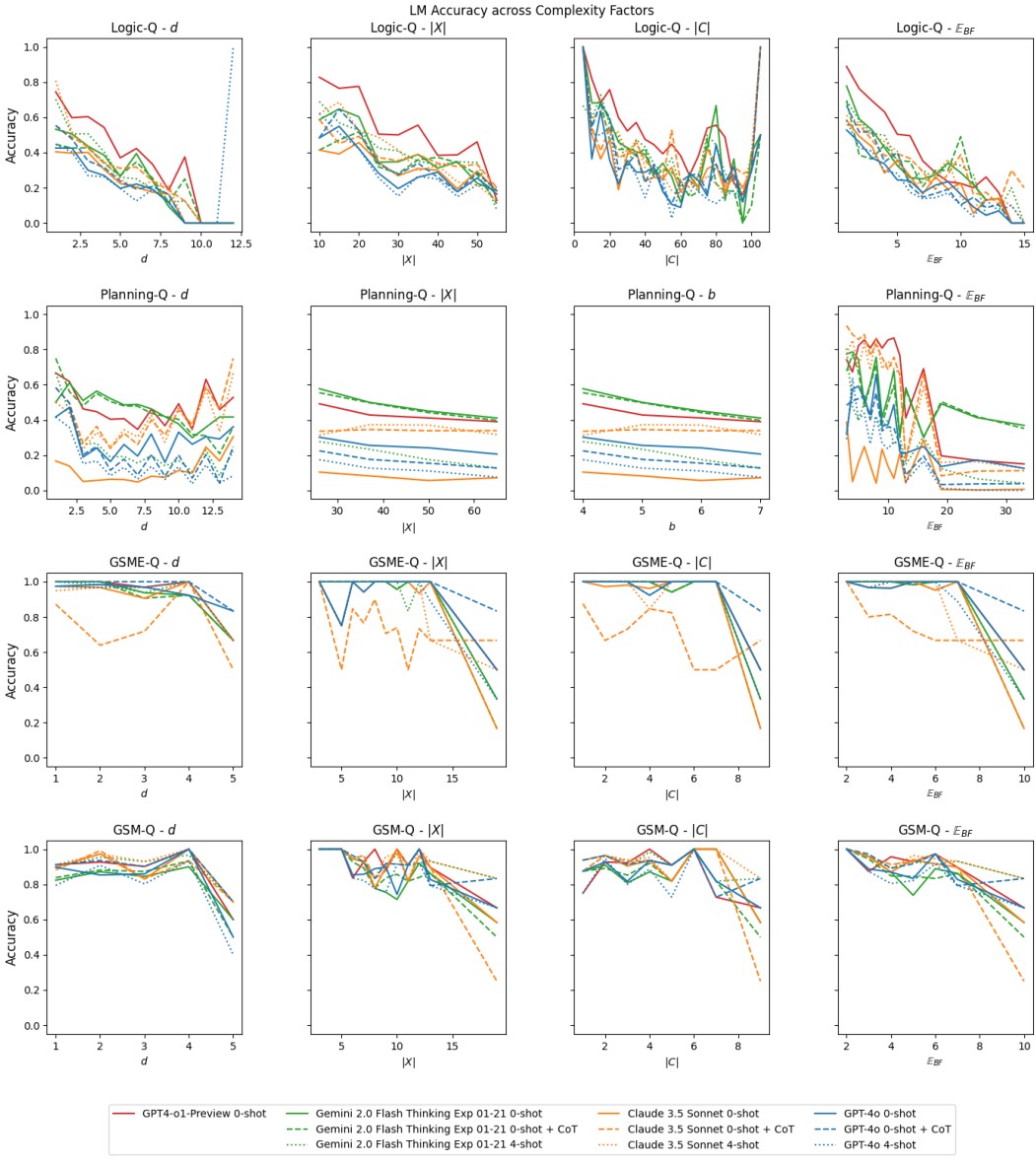

Figure 6: LM accuracies across varying backwards search depths $d$, number variables $|X|$, number constraints $|C|$, and expected number brute-force guesses $\mathbb{E}_{BF}$ for each domain, model, and prompt setting. To make the graph less noisy, we aggregate units of 5 on the $x$-axis for the Logic-Q setting for $|X|$ and $|C|$.

# H   Ablation: QUESTBENCH Accuracy Filtering For Perfect Reasoning

We report full accuracies on the ablation study described in Section 4.3 in  Table 7, and on the GSME-Q and GSM-Q domains in Table 8.  Similar to the Logic-Q domain, we found negligible difference between performance on the filtered version and the original QUESTBENCH for most model setups — though note that LMs are generally already able to saturate the original version of QUESTBENCH.

|  |  | Logic-Q | | | Planning-Q | | |
|---|---|---|---|---|---|---|---|
|  |  | Acc | Diff | # samples | Acc | Diff | # samples |
| ZS | Gemini 2.0 FTE | 37.4% | 0.0% | 1150 | 53.8% | **6.5%** | 4273 |
| | Claude 3.5 Sonnet | 30.8% | **0.7%** | 1120 | 11.3% | **3.8%** | 4368 |
| | GPT-4o | 27.4% | 0.0% | 1150 | 33.4% | **8.9%** | 2275 |
| | Gemini 1.5 Pro | 29.6% | **0.5%** | 1127 | 28.7% | **8.9%** | 2175 |
| | Gemini 1.5 Flash | 17.1% | **0.3%** | 1127 | 16.1% | **7.6%** | 1288 |
| | Gemma 2 2B IT | 0.1% | **0.0%** | 1101 | 0.0% | -10.8% | 1 |
| | Gemma 2 9B IT | 26.2% | **0.2%** | 1116 | 21.1% | **6.7%** | 95 |
| | Gemma 2 27B IT | 29.8% | **0.4%** | 1111 | 30.8% | **23.1%** | 117 |
| ZS + CoT | Gemini 2.0 FTE | 34.6% | 0.0% | 1150 | 52.3% | **6.1%** | 4318 |
| | Claude 3.5 Sonnet | 37.0% | **0.2%** | 1132 | 43.4% | **9.4%** | 5176 |
| | GPT-4o | 32.0% | **0.4%** | 1133 | 35.7% | **19.3%** | 1339 |
| | Gemini 1.5 Pro | 29.4% | **0.2%** | 1132 | 28.1% | **6.8%** | 2456 |
| | Gemini 1.5 Flash | 19.0% | **0.3%** | 1130 | 20.4% | **11.1%** | 1386 |
| | Gemma 2 2B IT | 0.3% | **0.0%** | 1088 | - | - | 0 |
| | Gemma 2 9B IT | 25.4% | **0.8%** | 1097 | 21.9% | **11.7%** | 73 |
| | Gemma 2 27B IT | 32.3% | **1.1%** | 1090 | 28.0% | **18.0%** | 75 |
| 4S | Gemini 2.0 FTE | 38.5% | **0.1%** | 1147 | 28.5% | **9.2%** | 4016 |
| | Claude 3.5 Sonnet | 36.3% | **0.3%** | 1131 | 44.7% | **10.1%** | 4264 |
| | GPT-4o | 25.5% | -0.1% | 1143 | 20.1% | **8.6%** | 2780 |
| | Gemini 1.5 Pro | 26.8% | **0.4%** | 1102 | 32.2% | **13.4%** | 2935 |
| | Gemini 1.5 Flash | 15.6% | **0.1%** | 1133 | 24.3% | **6.0%** | 3429 |
| | Gemma 2 2B IT | 25.1% | -0.0% | 1148 | 22.2% | **14.7%** | 9 |
| | Gemma 2 9B IT | 21.4% | -0.2% | 1135 | 11.7% | **6.6%** | 726 |
| | Gemma 2 27B IT | 26.4% | **0.1%** | 1130 | 22.7% | **12.0%** | 546 |

Table 7: Ablation: Language model accuracies at predicting the right question to ask on *only* filtered QUESTBENCH problems of which at least one well-specified version can be answered correctly by the model + prompt method. We use Gemini 2.0 FTE as a shorthand for Gemini Flash Thinking 2.0 Exp 01-21. See Section 4.3 for the experimental setup. **Acc** indicates accuracy on the filtered version of QUESTBENCH, **Diff** indicates difference between the accuracy in the filtered version vs. the original QUESTBENCH dataset (Table 1), and **# samples** indicates the number of samples in the filtered dataset, out of 1150 and 7500 examples in the original Logic-Q and Planning-Q, respectively. We **bold** all accuracy differences greater than zero.

| | | GSME-Q | | | GSM-Q | | |
|---|---|---|---|---|---|---|---|
| | | Acc | Diff | # samples | Acc | Diff | # samples |
| ZS | Gemini 2.0 FTE | 99.3% | **2.6%** | 145 | 84.5% | -0.3% | 283 |
| | Claude 3.5 Sonnet | 86.4% | -8.3% | 44 | 92.6% | **1.3%** | 175 |
| | GPT-4o | 97.5% | **0.8%** | 40 | 90.1% | **3.3%** | 111 |
| | Gemini 1.5 Pro | 58.5% | **10.1%** | 53 | 84.0% | **6.2%** | 50 |
| | Gemini 1.5 Flash | 96.2% | -0.5% | 26 | 66.7% | -5.2% | 18 |
| | Gemma 2 2B IT | 71.4% | **39.0%** | 7 | 45.8% | -3.8% | 24 |
| | Gemma 2 9B IT | 93.3% | **17.8%** | 15 | 40.4% | -0.6% | 52 |
| | Gemma 2 27B IT | 100.0% | **26.5%** | 25 | 85.7% | **14.5%** | 7 |
| ZS + CoT | Gemini 2.0 FTE | 99.3% | **3.3%** | 145 | 86.2% | **0.1%** | 283 |
| | Claude 3.5 Sonnet | 74.2% | 0.0% | 151 | 91.0% | 0.0% | 288 |
| | GPT-4o | 99.3% | 0.0% | 151 | 91.7% | -0.3% | 277 |
| | Gemini 1.5 Pro | 99.3% | **2.6%** | 145 | 79.2% | **0.0%** | 274 |
| | Gemini 1.5 Flash | 98.7% | 0.0% | 151 | 75.6% | -1.2% | 266 |
| | Gemma 2 2B IT | 61.0% | **4.1%** | 118 | 70.1% | **1.7%** | 157 |
| | Gemma 2 9B IT | 57.0% | **0.1%** | 114 | 39.3% | **2.8%** | 224 |
| | Gemma 2 27B IT | 82.1% | **1.3%** | 134 | 81.7% | **2.2%** | 224 |
| 4S | Gemini 2.0 FTE | 96.7% | 0.0% | 151 | 92.6% | -0.2% | 282 |
| | Claude 3.5 Sonnet | 92.0% | -3.3% | 88 | 95.1% | **1.0%** | 204 |
| | GPT-4o | 96.6% | **0.6%** | 59 | 84.0% | **0.0%** | 188 |
| | Gemini 1.5 Pro | 85.2% | -6.8% | 61 | 81.1% | -4.4% | 132 |
| | Gemini 1.5 Flash | 94.1% | -1.2% | 34 | 78.1% | -2.8% | 137 |
| | Gemma 2 2B IT | 95.0% | **4.3%** | 20 | 37.6% | **0.1%** | 125 |
| | Gemma 2 9B IT | 76.5% | -10.3% | 34 | 63.1% | **7.5%** | 157 |
| | Gemma 2 27B IT | 97.7% | **1.0%** | 44 | 64.7% | -1.9% | 190 |

Table 8: Ablation: Language model accuracies at predicting the right question to ask on *only* filtered QUESTBENCH problems whose well-specified versions can be answered correctly at least once by the model + prompt method. We use Gemini 2.0 FTE as a shorthand for Gemini Flash Thinking 2.0 Exp 01-21. See Section 4.3 for the experimental setup. **Acc** indicates accuracy on the filtered version of QUESTBENCH, **Diff** indicates difference between the accuracy in the filtered version vs. the original QUESTBENCH dataset, and **# samples** indicates the number of samples in the filtered dataset, out of 151 representative examples in GSME-Q and 288 representative examples in GSM-Q. We **bold** all accuracy differences greater than zero.

|  |  | Logic-Q | Planning-Q | GSME-Q | GSM-Q |
|---|---|---|---|---|---|
| ZS | Gemini Flash Thinking 2.0 Exp 01-21 | 60.87 | 0.82 | 22.35 | 36.83 |
|  | Claude 3.5 Sonnet | 57.09 | 10.77 | 65.22 | 59.85 |
|  | GPT-4o | 36.26 | 1.23 | 30.27 | 60.87 |
|  | Gemini 1.5 Pro | 47.44 | 9.88 | 12.42 | 55.28 |
|  | Gemini 1.5 Flash | 34.03 | 0.00 | 39.41 | 51.16 |
|  | Gemma 2 2B IT | 47.03 | 0.00 | 2.56 | 4.08 |
|  | Gemma 2 9B IT | 48.39 | 0.00 | 40.66 | 74.67 |
|  | Gemma 2 27B IT | 50.29 | 0.10 | 6.13 | 56.93 |
| ZS + CoT | Gemini Flash Thinking 2.0 Exp 01-21 | 61.70 | 0.89 | 12.42 | 28.57 |
|  | Claude 3.5 Sonnet | 58.87 | 15.45 | 96.58 | 68.49 |
|  | GPT-4o | 45.95 | 0.37 | 83.70 | 62.86 |
|  | Gemini 1.5 Pro | 53.37 | 4.15 | 88.15 | 65.11 |
|  | Gemini 1.5 Flash | 55.71 | 4.53 | 90.59 | 69.09 |
|  | Gemma 2 2B IT | 35.96 | 0.05 | 53.40 | 21.67 |
|  | Gemma 2 9B IT | 48.98 | 0.00 | 64.91 | 67.12 |
|  | Gemma 2 27B IT | 47.87 | 0.74 | 91.37 | 60.19 |
| 4S | Gemini Flash Thinking 2.0 Exp 01-21 | 57.95 | 5.42 | 74.90 | 50.00 |
|  | Claude 3.5 Sonnet | 51.65 | 31.52 | 91.37 | 86.41 |
|  | GPT-4o | 24.49 | 13.39 | 40.21 | 82.69 |
|  | Gemini 1.5 Pro | 32.35 | 9.49 | 66.37 | 78.74 |
|  | Gemini 1.5 Flash | 27.62 | 20.82 | 75.21 | 75.95 |
|  | Gemma 2 2B IT | 37.95 | 46.52 | 15.03 | 61.75 |
|  | Gemma 2 9B IT | 43.75 | 2.85 | 16.00 | 70.62 |
|  | Gemma 2 27B IT | 36.04 | 9.06 | 15.20 | 75.14 |
| Random | (proportional) | 41.97 | 41.75 | 42.90 | 50.0 |

Table 9: F1 scores for identifying which problems are underspecified in QUESTBENCH. We include a random baseline where a randomly-sampled, proportional number of examples are labeled "not sure". We found that most LLMs have difficulty assessing when problems are underspecified, particularly in difficult domains like Planning-Q.

# I Computational Resources

Generating the dataset for the Logic-Q and Planning-Q mostly requires CPU resources to conduct the backwards search. The search and filtering stages for creating each domain required a few (between three to seven) days each. For experiments, we queried Gemini, GPT-4o, and Claude models through the API. We launched Gemma models on either TPU v5e node pools, or 1-2 H100 GPUs, and ran inference with them.

Evaluating each model took a few hours for Logic-Q, up to a few days for Planning-Q, and a few minutes for GSM-Q / GSME-Q.

# J Limitations

Our work includes a few key assumptions that may limit its generalizability and practicality. For example, QUESTBENCH focuses exclusively on 1-sufficient CSPs, isolating the minimal case of information acquisition. However, real-world problems often include multiple missing pieces of information or more complex dependency structures. QUESTBENCH represents a first attempt at dealing with underspecification, favoring formalism and ease of evaluation over generality. Furthermore, our domains were chosen for their clear CSP formalizations, but may not capture the full range of naturalistic language understanding and ambiguity found in everyday tasks. Finally, using multiple choice for question selection simplifies evaluation and ensures consistency, but may overestimate models' performance compared to open-ended question generation.

# K Broader Impacts

By highlighting gaps in current LLMs' ability to ask clarification questions, this work could guide the development of more interactive, trustworthy AI systems. Better information-seeking behavior is critical for high-stakes domains such as education, healthcare, and scientific research, where clarifying underspecified inputs can reduce misunderstandings and improve decision-making. Moreover, our benchmark offers a well-defined setting for studying model uncertainty and epistemic humility, which are increasingly important for alignment and safety. However, improved question-asking abilities could be misused in automated persuasion, e.g., models that dynamically elicit personal information through carefully crafted queries. While QuestBench does not train models to ask open-ended questions, the underlying capabilities evaluated could inform systems with stronger elicitation power. Coupling queries with rationales for the model's query may allow user greater control over the information shared.

