# OpenReview forum: "QuestBench: Can LLMs ask the right question to acquire information in reasoning tasks?"
_NeurIPS.cc/2025/Datasets_and_Benchmarks_Track — NeurIPS 2025 Datasets and Benchmarks Track poster_

### Official Review · Reviewer_nPYJ · 2025-06-13

**Rating:** 5
**Confidence:** 4

**Summary:**

This paper considers underspecified queries that are only solvable by acquiring one additional piece of information, e.g., the value of one variable. A dataset covering logical reasoning tasks, PDDL planning problems, and math problems is synthesized to provide such 1-sufficient CSPs. State-of-the-art LLMs are evaluated on the dataset.

**Additional Feedback:**

Minor issues:

1. Try not to say both LLM and LM in the paper.

2. Note the irregular line break at the beginning of Page 2.

**Dataset Code Accessibility:**

Yes

**Dataset Code Comments:**

I checked Hugging Face and GitHub. Data and code are available and documented.

**Ethical Comments:**

I did not see any ethical issues. The dataset was adpated from existing public datasets.

**Ethical Considerations:**

No, there are no or only very minor ethics concerns

**Final Justification:**

My concerns have been mostly addressed. I have increased my rating. Still, the presentation requires a significant revision to avoid misunderstandings.

**Limitations Weaknesses:**

W1: The dataset contains four subsets. Planning-Q seems too hard to be useful for evaluating LMs, which have to perform random guess as indicated in Section 4.2. GSM-Q and GSME-Q, on the other hand, are too simple to be useful for evaluating current LLMs. Logic-Q becomes the only useful subset, which, however, is also the smallest one containing only 1,150 questions.

W2: For a 1-sufficient CSP, there may be more than one sufficient variable, that is, the gold-standard question choice may not be unique. It is unclear whether this has been considered during dataset construction. There is some related disucssion in B.1.2 about sufficient variables at different reasoning depths, but what if there are multiple sufficient variables at the same depth?

W3: It is unclear why only 288/151 representative tasks from GSM-Q/GSME-Q were used for evaluation. How did you guarantee representativeness?

**Strengths Contributions:**

S1: Interesting and well-motivated problem setting.

S2: Generally solid benchmark design.

S3: Extensive experiments.

---

> ### Author Rebuttal · Authors · 2025-07-30
>
> Thank you for your insightful feedback! Your suggestions will greatly strengthen our paper. Below, we address the weaknesses that you raised.
>
> ## W1: The dataset contains four subsets. Planning-Q seems too hard (section 4.2 suggests LLMs are performing random guess), GSM-Q and GSME-Q are too simple. Logic-Q is the only useful subset, but is also the smallest one containing only 1,150 questions.
>
> We would like to gently push back against Planning-Q being too complex. We see that the most recent thinking models, such as o1-preview and Gemini-Flash-Thinking are able to perform nontrivially on Planning-Q, achieving ~40-50%. There are strong indications that thinking models are improving rapidly on this benchmark, as the gap between thinking and non-thinking models is over 20%. If this scaling curve holds, then we may expect rapid improvement on this domain.
>
> While Section 4.2 suggests that current LLMs’ performance on Planning-Q is strongly negatively correlated with $\mathbb{E}_{BF}$, which is consistent with validating one choice at a time, this is a valid reasoning approach and may be more efficient than breadth-first search for some problems. We do not see this as necessarily evidence that LLMs are doing the “wrong thing”.
>
> Finally, we believe our benchmark captures the crux of the difficulty of real-world planning in the presence of ambiguity. Being unable to perform well on our benchmark means LLMs will have difficulty with real-world planning tasks that often contain far more variables and far more underspecifications.
>
> ## W2: For a 1-sufficient CSP, there may be more than one sufficient variable, that is, the gold-standard question choice may not be unique
>
> This is a good point – indeed, there may be multiple valid sufficient variables for any 1-sufficient CSPs. We considered this when building the benchmark, and the benchmark accepts multiple valid answers. We say that if the LLM picks /any/ 1-sufficient variable, it is correct (see footnote 3 at the bottom of page 3). We will further clarify this in future versions of the paper!
>
> ## W3: It is unclear why only 288/151 representative tasks from GSM-Q/GSME-Q were used for evaluation. How did you guarantee representativeness?
>
> Great question! Because we were constrained by time and cost, we decided to focus our evaluation budget on the more difficult domains to test the limit of the models. Thus, we selected a small subset of these two domains -- mainly to demonstrate the point that modern LLMs saturate these domains -- and focus the rest of our analysis mainly on the other two domains. However, as a sanity check, we report GPT-4o (ZS) evaluations on the full datasets below:
>
> > GSME-Q: 99.39%
> >
> > GSM-Q: 84.05%
>
> These results affirm our claim that performance saturates on these subsets, consistent with our findings on the smaller subset in the paper.
>
> ## Minor issue 1: Try not to say both LLM and LM in the paper.
>
> Good point! We’ll revise the future versions of the paper to reflect these changes
>
> ## Minor issue 2: Note the irregular line break at the beginning of Page 2.
> Great catch – we’ll fix this in future versions of the paper.

---

> > ### Comment · Reviewer_nPYJ · 2025-08-03
> >
> > Thanks for your reply. Most of my concerns have been addressed.
> >
> > Still, I am confused by the following potential contradiction: In your reply you said "o1-preview and Gemini-Flash-Thinking are able to perform nontrivially on Planning-Q", but Seciton 4.2 concludes that "those LLMs might be randomly guessing, or verifying one choice at a time". Do you think "verifying one choice at a time" is trivial?

---

> > > ### Author Response · Authors · 2025-08-04
> > >
> > > Good question! To clarify, "verifying one choice at a time" does not always have to be trivial -- for example, you can think of brute-force search as a version of "verifying one choice at a time". Particularly in Planning-Q, verification doesn't just require a single check, but can entail rolling out entire trajectories -- potentially from multiple possible initial states. We’d also like to clarify that our analyses in Section 4.2 are intended as behavioral analogies, not claims about the actual mechanisms used by LLMs. While we find evidence against the specific formulation of backwards search or tree search described in the paper, this does not imply the model is following any particular algorithm. Our goal is to interpret model behavior through the lens of these baselines, while acknowledging that the true underlying process may be far more complex or entirely different. We will clarify what we mean by "verifying one choice at a time" in Section 4.2!
> > >
> > > In the rebuttal, by "o1-preview and Gemini-Flash-Thinking are able to perform nontrivially on Planning-Q", we just meant o1-preview and Gemini-Flash-Thinking have relatively good performance on Planning-Q compared to other non-thinking models. We agree the original wording in the rebuttal could be clearer!

---

> > > > ### Comment · Reviewer_nPYJ · 2025-08-06
> > > >
> > > > Thank you for your clarification. I am generally satisfied after discussion. Will increase the rating. I hope the final version can be better presented, in particular in terms of wording and consistency.

---

### Official Review · Reviewer_w14e · 2025-06-27

**Rating:** 5
**Confidence:** 3

**Summary:**

In this manuscript, the authors proposed QuestBench, a multiple-choice question benchmark set for evaluating the ability of LLMs to ask appropriate clarification questions to solve underspecified reasoning tasks. Technically, the authors formalize the problem as a constraint satisfaction problem (CSP) with missing variable assignments. They then construct four types of benchmarks, including Logic-Q (logical reasoning), Planning-Q (planning problems), GSM-Q (grad school math problems) and GSME-Q (equation version of GSM-Q). The authors evaluated LLMs on QuestBench and observed limited performance of LLMs on Logic-Q and Planning-Q. They revealed through analysis that existing models lack the ability to identify appropriate clarification questions for underspecfied tasks even though they are capable for solving the fully specified version.

**Dataset Code Accessibility:**

Yes

**Dataset Code Comments:**

The authors provided the code of the benchmark on github and released the dataset on huggingface.

**Ethical Considerations:**

No, there are no or only very minor ethics concerns

**Final Justification:**

The paper is techically solid and well-written. My concerns in review have been well addressed by the authors' rebuttal. I would like to recommend accept the paper with score = 5.

**Limitations Weaknesses:**

1. A lack of integration with real-world science reasoning tasks. Logical and mathematical reasoning is common in science subjects (e.g., MMMU (Science), ScienceQA, etc). However, this is less discussed in the manuscript. Could the authors explain the potential of QuestBench to integrate with science subjects (e.g., automatically generating clarification benchmark based on existing reasoning benchmarks)?
2. Simpler clarification than real-world scenarios. As the authors mentioned in the appendix, the QuestBench only covers 1-sufficient CSPs and are multiple-choice questions, which is simpler than real-world underspecified reasoning tasks, which could miss more than one variables and requires open-ended asking of clarification questions.
3. The authors mentioned in page 6 that “We suspect that this is due to these domains having a smaller number of variables and constraints, and requiring shallower search depth than the other two domains” -- Is there any evidence to support this assumption?

**Strengths Contributions:**

1. The evaluation of the ability to ask clarification questions to solve underspecified reasoning tasks is interesting and novel. The authors gave a clear demonstration of the problem and its potential value in real-world reasoning tasks for LLMs.
2. The paper is well written, with clear formulation and evaluation analysis. The authors provided a clear formulation of CSP. In addition, the experiment is comprehensive and easy to understand.
3. Experimental findings are insightful. The authors revealed significant ability gap between logical clarification and mathematical clarifiation, posing importance for enhance the logical and planning ability of LLMs. Further analysis in Section 4.3 revealed that correctly respond to fully specifed problem is insuffcient for asking clarification questions in underspecified version. This emphasizes the necessity to enhance the ability of LLMs to actively interact with the envrionment and collect missing informaitions to solve underspecfied reasoning tasks in real-world scenarios.

---

> ### Author Rebuttal · Authors · 2025-07-30
>
> Thank you for your thoughtful review! Your suggestions will greatly strengthen our paper. Below, we address several weaknesses that you have raised:
>
> ## W1: Lack of integration with real-world science reasoning tasks. Explain potential of QuestBench to integrate with science subjects (e.g., automatically generating clarification benchmark based on existing reasoning benchmarks)?
>
> Please see our response to W3 for reviewer LTGU, where we go through an example of a ScienceQA question, and how it can integrate with our framework! Overall, we expect the data generation pipeline in this case to be similar to how we generated GSM-Q.
>
> ## W2: Simpler clarification than real-world scenarios. 1-sufficient CSPs could miss more than one variables and requires open-ended asking of clarification questions.
>
> Please see our response to W1 for reviewer LTGU!
>
> ## W3: Problems are and are multiple-choice questions, which is simpler than real-world underspecified reasoning tasks
>
> Please see our response to W2 for reviewer LTGU!
>
> ## W4: “We suspect that this is due to these domains having a smaller number of variables and constraints, and requiring shallower search depth than the other two domains” -- Is there any evidence to support this assumption?
>
> Yes! We drew this conclusion based on Fig. 6, we know the LM accuracy is generally negatively correlated with number of variables and depth. In the GSM domain, we can see that the model is almost perfectly correct until depth exceeds 4 and number of variables exceeds 12. We suspect that model accuracy would continue to trend downward if we increased depth and number of variables. We will clarify this in future versions of the paper.

---

> > ### Comment · Reviewer_w14e · 2025-08-04
> > **Reviewer Response**
> >
> > Thank you for the detailed rebuttal. My concerns have been well addressed!
> >
> > Please remember to mention Figure 6 when describing your assumption in the revised version.

---

### Official Review · Reviewer_LTGU · 2025-06-29

**Rating:** 4
**Confidence:** 3

**Summary:**

This paper explores whether LLMs can effectively identify and request the minimal necessary information to resolve underspecified reasoning tasks. It formalizes the task as an underspecified Constraint Satisfaction Problem (CSP), specifically focusing on cases that require asking only one additional clarifying question ("1-sufficient CSPs").

To investigate this problem, this paper introduces QuestBemch, a benchmark to test LLMs' capabilities in identifying and asking critical questions across four domains: (1) Logic-Q: logical reasoning tasks with one missing proposition, (2) Planning-Q: PDDL planning problems with partially-observed initial states, (3) GSM-Q: human-annotated grade school math problems with one missing variable, and (4) GSME-Q: equation-based version of GSM-Q.

Models such as GPT-4o, Claude 3.5, and Gemini variants were tested. The authors found that while LLMs excelled on simpler math tasks (over 80% accuracy on GSM-Q and GSME-Q), they struggled significantly with logical reasoning and planning (below 50% accuracy on Logic-Q and Planning-Q).

**Dataset Code Accessibility:**

Yes

**Dataset Code Comments:**

The code looks good to me.

**Ethical Considerations:**

No, there are no or only very minor ethics concerns

**Final Justification:**

Most of my concerns are addressed. I still recommend acceptance.

**Limitations Weaknesses:**

- The paper exclusively examines "1-sufficient" CSPs, where only one clarifying question is needed. This highly specific restriction severely limits the generalizability of the findings, because real-world scenarios typically involve multiple interacting sources of underspecification.
- The choice-based framework significantly simplifies the question-asking task. Instead of generating novel questions, models merely choose from pre-defined options. This design likely inflates measured performance, masking limitations in models' generative abilities. A more realistic evaluation would have included free-form or open-ended question generation, which would reflect practical challenges.
- While the benchmark covers logic, planning, and mathematics, these domains are artificially simplified and structured. For example, the "Blocks World" (Planning-Q) and logical reasoning tasks (Logic-Q) are classical problems that have limited resemblance to the complexity and ambiguity encountered in real-world scenarios.

**Strengths Contributions:**

- This paper formalizes the capability of question-asking in reasoning tasks as an underspecified Constraint Satisfaction Problem (CSP). This formalization seems novel and rigorous. By distinguishing between semantic ambiguity (multiple interpretations exist) and underspecification (information is objectively missing), the paper offers a clear and precise benchmark.
- QuestBench covers a few distinct reasoning domains: logic, planning, and mathematics, both verbal and symbolic, and it also includes both human-annotated and algorithmically generated tasks, ensuring diverse distribtuion and wide coverage.
- This paper doesn't just report accuracy; it digs deeper into factors influencing LLM performance, such as the complexity and depth of reasoning required. By analyzing correlations between LLM performance and task difficulty metrics (variables, constraints, search depth), it provides valuable insights into the types of reasoning mechanisms current models utilize or fail to employ effectively.

---

> ### Author Rebuttal · Authors · 2025-07-30
>
> Thank you for your thoughtful feedback! Your suggestions will greatly strengthen our paper. Below, we address several weaknesses that you have raised:
>
> ## W1: "1-sufficient" CSPs limits the generalizability of the findings
> We agree that real-world reasoning tasks often require acquiring multiple pieces of missing information, and thus go beyond the 1-sufficient setting. However, we chose to focus on 1-sufficient CSPs in this initial benchmark for three main reasons:
> 1. Foundational nature of 1-sufficient CSPs:
>  As we note in the paper (top of page 4), 1-sufficient CSPs are a fundamental building block for more complex k-sufficient problems. Any k-sufficient CSP can become 1-sufficient after k−1 clarifying questions. Thus, performance on 1-sufficient CSPs provides a necessary (if not sufficient) indicator of a model’s capacity for information acquisition. In fact, we can prove that accuracy on 1-sufficient problems provides an upper bound for performance on harder k-sufficient cases.
>
> For any k-sufficient question-asking problem, assume the accuracy for selecting the right questions is $a_k$, and the sequence of questions is $q_{k}, q_{k-1}, …, q_1$. We have:
>
> $$\mathbb{E}\left[a_k\right] = \mathbb{E}\left[p( [q_k, q_{k-1}, …, q_1] \in C_k  \mid \text{k-sufficient problem} )\right]$$
> $$= \mathbb{E}\left[p(q_1 \in C_1 \mid  \text{1-sufficient problem} ) p(q_k, …, q_{2} \mid  \exists q_1 s.t. [q_k, …, q_{2}, q_{1}] \in C_k \right]$$
> $$< \mathbb{E}\left[ p(q_1 \in C_1 \mid  \text{1-sufficient problem} \right] = a_1$$
>
> Here $C_k$ is the set of all correct k question tuples. This shows we always have $a_k < a_1$.
>
> 2. Evaluation tractability and clarity:
> Rigorous evaluation of information-seeking behavior becomes increasingly ambiguous as k increases. While even 1-sufficient CSPs may admit multiple sufficient variables, each such variable on its own suffices to solve the task. This allows us to frame the benchmark as a multiple-choice problem where one or more answers can be accepted as correct, and correctness can be automatically verified. In contrast, k-sufficient problems often require reasoning about joint variable sets, where constructing natural and unambiguous question choices can be harder — e.g., “What is the the value of variables x₁, x₂, ..., x_k?” may not correspond to any intuitive or useful question in natural language.
> 3. Complexity of generalizing beyond 1-sufficiency:
>  While one could theoretically reduce a k-sufficient CSP to a 1-sufficient one by introducing a “conjunctive variable” representing all k unknowns, this construct rarely arises naturally from the structure of the original problem and adds cognitive and computational complexity. Furthermore, not all multi-variable queries may be equally “simple” or interpretable. For example, asking about “Alice’s favorite color and shape” may be simpler than asking about “Alice’s favorite color and Bob’s favorite shape”. A multiple-choice evaluation of k-sufficient CSPs will not be able to account for these complexities.
> Finally, we note that our benchmark includes significant variation in problem difficulty, even within the 1-sufficient setting (see Section 2.3 and Table 5). This allows us to probe LLM behavior along axes like search depth, variable count, and expected guess complexity, which shed light on model limitations that are likely to persist or even amplify in k-sufficient settings.
>
>
> We will include the aforementioned discussion in future versions of the paper!
>
> ## W2: Choice-based framework simplifies the question-asking task, likely inflating measured performance. Free-form or open-ended question generation would reflect practical challenges.
>
> We agree that multiple-choice options are rarely available in real-world interactions. However, we chose this format for several important reasons:
> 1. Automated and rigorous evaluation: Multiple-choice framing allows for scalable, objective evaluation of clarification question quality. Alternatives like reward models or user simulators introduce new challenges, such as simulator brittleness or vulnerability to reward hacking, and often require extensive human effort to validate.
>
> 2. A necessary precursor to open-ended performance: Even when presented with answer choices, current LLMs frequently fail to select a correct clarifying question. This suggests that open-ended question-asking in complex reasoning problems—where no guidance is given—would only be harder. We view strong multiple-choice performance as a necessary (though not sufficient) precondition for progress toward open-ended systems.
>
> 3. The presence of options does not always make the problem simpler. For example, in problems for which search depth is low and the number options is high, it makes more sense to run breadth-first search than brute-force-search, for which the difficulty doesn’t depend on the presence or number of options.
>
> 4. Benchmark extensibility to open-ended evaluation: Our benchmark can be readily used in open-ended settings by simply omitting the multiple-choice options. Indeed, in Section 4.3, we evaluated LLMs' ability to detect whether a problem is underspecified without presenting answer choices. We found that models frequently failed to recognize underspecification—a prerequisite for asking a clarifying question in an open-ended interaction.
>
> Our main findings and subsidiary analyses in Section 4.3 point to substantial current limitations in both (1) recognizing the need to ask a question, and (2) identifying what that question should be. As such, we believe QuestBench's multiple-choice format remains a valuable diagnostic tool for progress on both fronts. We plan to add this discussion in future versions of the paper.
>
> ## W3: Domains are artificially simplified and structured, have limited resemblance to the complexity and ambiguity encountered in real-world scenarios.
>
> Our benchmark was deliberately designed to prioritize **rigorous, automatable evaluation** over coverage of inherently fuzzy or ambiguous real-world scenarios (see Abstract and top of page 1). We believe performing well on our well-defined benchmarks is a prerequisite for generalizing to more fuzzy, real-world reasoning applications. Since current SOTA models fail to saturate even our well-defined settings, we argue they are not yet ready for more complex, real-world underspecification.
>
> As discussed in Section 2.1, we explicitly distinguish semantic ambiguity (multiple valid interpretations) from underspecification (insufficient information for resolution). Prior work has studied ambiguity extensively, but underspecification remains relatively underexplored. By focusing on domains where ambiguity is minimized and the missing information is well-defined, we can cleanly isolate models’ ability to detect and acquire missing information.
>
> That said, our constraint satisfaction problem (CSP) framework is not restricted to the domains in QuestBench. It extends naturally to a variety of real-world tasks. For example, consider this science reasoning problem from ScienceQA:
>
> > In a group of guppies, some individuals have a spotted tail and others have an unspotted tail. In this group, the gene for the tail spots trait has two alleles. The allele for a spotted tail (I) is dominant over the allele for an unspotted tail (i). Sebastian is a guppy from this group. Sebastian has the homozygous genotype ii for the tail spots gene.
> > Based on this information, what is Sebastian's phenotype for the tail spots trait?
>
> This task can be broken down into a CSP as follows:
>
> * Variables & Domains
> > G = Sebastian’s genotype ∈ {II, Ii, ii}
> >
> > P = Sebastian’s phenotype  ∈ {Spotted, Unspotted}
>
> * Constraints
> > G ∈ {II, Ii} → P = Spotted [Dominance rule]
> >
> > G ∈ {ii} → P = Unspotted [Recessive expression]
> >
> > G = ii [Given]
>
> * Target Variable: P
>
> Using this CSP, the target (Sebastian’s phenotype) can be logically inferred. Moreover, as in GSM-Q, one can introduce underspecification by withholding the genotype and evaluating whether the model can identify that asking for G is necessary to resolve P.
>
> In short, the CSP framework generalizes well to other structured reasoning settings (e.g., science, commonsense, causality), creates a general strategy (including both automatic approaches and re-usable human data collection instructions) for constructing new information-seeking datasets, and could provide a foundation for future benchmarks that push closer toward real-world underspecification while maintaining evaluability.

---

> > ### Comment · Reviewer_LTGU · 2025-08-05
> >
> > Thanks to the authors for their response. I don't have any further questions.

---

### Official Review · Reviewer_3KGM · 2025-07-07

**Rating:** 5
**Confidence:** 3

**Summary:**

The paper proposes and releases QuestBench, a benchmark aimed at evaluating the ability of LLMs and LRMs to identify and ask about missing information needed to solve a logical/planning/math task. The authors study this by formalizing several benchmarks from prior work as CSPs with missing variable assignments, and conduct a study on a set of models, summarizing the insights.

**Additional Feedback:**

- When reading the paper, the first instantiation that may fit the CSP formulation that came to mind are logical puzzles (einstein, sudoku, mazes). Do you think it is viable to extend QuestBench to those?
- How do you think the results would change for 2-sufficient problems? Intuitively, it seems much more difficult due to exponential growth of possible missing tuples, but perhaps the authors have a better insight.

**Dataset Code Accessibility:**

Yes

**Dataset Code Comments:**

The code and the data are accessible at the given link and well-documented.

**Ethical Considerations:**

No, there are no or only very minor ethics concerns

**Final Justification:**

After reading the author's rebuttals and other reviews I remain positive about the paper. The limitations are clear and stated upfront and I believe that the paper is a good contribution regardless, so I keep my score at 5.

**Limitations Weaknesses:**

While the authors recognize this, and I believe the paper is a solid contribution even despite this, the multiple choice formulation may limit the scope of the conclusions that can be made. My understanding is that the brute-force solver does not need necessarily to check for all variables but only those mentioned in the 4 given answer choices, to determine what information is exactly missing? It would be interesting to know if this happens in practice.

Independently, I believe the paper would be stronger with a qualitative study of reasoning traces. For example for Gemini FTE I would be interested to know how much bruteforcing/search is employed by the model and if there are common failure modes.

**Strengths Contributions:**

- The work is quite timely as reasoning is one of the main foci of current LLM research. However, most such work focuses on the ability of LLMs to solve tasks via reasoning but less so on the issue of gathering missing information. The latest design of popular AI endpoints (deep research, image generation) includes follow-up questions before executing the user request which implies that requests in the wild are often underspecified and that the ability to reason about this and gather missing info is an important skill.
- The approach is reasonable and App. A makes a convincing case that it is novel (although I am not deeply familiar with benchmarks on this topic). In particular the distinction between semantic ambiguity vs underspecification is interesting, and I recognize the value of having a well-formalized phrasing of the problem, which the authors achieve by casting existing benchmarks as CSPs. This has an additional value as the community is already familiar with the source benchmarks and their pros/cons so it is easier to adapt to QuestBench.
- The effort is thorough. The benchmark includes several very different areas (math, propositional logic, planning), and the evaluation is detailed and contains many interesting non-trivial takeaways. First, it is interesting that on planning/reasoning the gap between solving the task and identifying the missing variable exists, and is much larger than for math tasks. Second, I appreciate that the authors took significant effort to relate this to axes of difficulty of problems. The authors also include options where the problem is in fact specified ("no questions needed"), different prompting strategies (cot/fewshot), the analysis of brute-force solutions, and the additional study that asks if the models can recognize uncertainty themselves or overconfidently guess the answer. All these are questions that I had reading the paper and provide valuable insight that complements the main results.
- Paper is well written and a pleasure to read. The benchmarks from prior work are concisely introduced and it is clearly explained how they are adapted to make QuestBench. I appreciate the example in Sec. 2 which makes it immediately clear what is the focus of the bencmark.
- The release is nicely structured and well-documented (github code and huggingface dataset).

---

> ### Author Rebuttal · Authors · 2025-07-30
>
> Thank you for your insightful review! Your suggestions will greatly strengthen our paper. Below, we address several weaknesses that you have raised:
>
> ## W1: Multiple choice formulation may limit the scope of the conclusions that can be made.
>
> Please see our response to W2 for reviewer LTGU!
>
> ## W2: Does the brute-force solver check for all variables or only those mentioned in the 4 given answer choices?
>
> We check for *all* variables, and there may be more than 4 answer choices (see Table 5 for breakdown of average # of choices in each domain.) We say that if the LLM picks /any/ 1-sufficient variable, it is correct. The dataset has multiple correct answers. We will further clarify this in future versions of the paper!
>
> ## W3: Qualitative study of reasoning traces.
>
> Great suggestion! While Gemini 2.0 Flash Thinking Experimental is no longer available, we examined some reasoning traces from Gemini 2.5 Flash on a Logic-Q and Planning-Q problem, respectively, and record the results of this analysis below. (Note we are not allowed to link to the traces in this rebuttal.)
>
> * Logic-Q prompt: we use the first example in the Logic-Q subset of Questbench, which has goal variable “tired”. The ground-truth questions here include “different”, “frightened” and “hurt”
> We examine the Gemini 2.5 Flash reasoning trace on this problem (full trace excluded for character limit). The model settles on an incorrect question: “Is Alice precious”. Based on the reasoning traces, it looks like the model begins by pursuing a backwards breadth-first search approach, first “exhaust[ing] all rules containing "tired”, identifying “different" and "shiny," or "difficult" and "precious" as the “required attributes”. It concludes that it needs to establish "different" and "shiny," or "difficult" and "precious." It then switches to a forward search for each of these elements from the known facts. It appears that while it thoroughly investigates the positive direction (what is needed to establish that Alice /is/ tired), it never investigates the negative direction (what is needed to establish that Alice is /not/ tired). Thus it never establishes that “precious” is the *only* question it needs to ask.
>
> * Planning-Q prompt: we use the sixth example in the Planning-Q subset of Questbench, which has initial conditions “(on a c); (ontable b); (clear a); (holding d); (on c e); not (clear b)” and ground-truth question “No questions needed”. We examine the Gemini 2.5 Flash reasoning trace on this problem (full trace excluded for character limit). The model appears to be using a non-search-based approach of /visualizing/ the state of the table and the stacks of blocks, which may explain why LLM performance only weakly correlated with search complexity factors in this domain (see Section 4.2). Here, the model settles on an incorrect question: “Is (clear b)?” (despite (clear b) already being established as false), because it appears to have never made the connection that e could be on b, and incorrectly asserts there is a contradiction because no block could be on b.
>
> This is a great suggestion and we will include these traces and analyses of them in subsequent versions of the paper.
>
> ## Q1: The first instantiation that may fit the CSP formulation that came to mind are logical puzzles (einstein, sudoku, mazes). Do you think it is viable to extend QuestBench to those?
> These are all great domains to consider! We agree that domains like Sudoku, Einstein’s puzzle, and mazes can naturally fit the CSP framework and are viable extensions to QuestBench. For example, given a well-specified Sodoku puzzle, we can mask out one of the known cells, and see if the LM can figure out the masked cell that it needs to ask about. Similar to the domains in the paper, the main challenge lies in ensuring that a single missing element makes the problem 1-sufficient and in defining precise clarification questions. We see this as a promising direction for future work.
>
> Please also see our response to W1 for reviewer w14e for an extension to a real-world science question!
>
> ## Q2: How do you think the results would change for 2-sufficient problems?
>
> This is a great question—thank you for raising it. One reference point is the complexity of search algorithms. In Logic-Q, where LLMs exhibit behavior roughly consistent with search, we expect performance to degrade for 2-sufficient problems.
>
> Specifically, brute-force search scales combinatorially with $k$, as the model would need to evaluate all $\binom{n}{k}$ combinations of variable subsets. In contrast, the worst-case complexity of backwards breadth-first search remains the same between 1-sufficient and $k$-sufficient problems, since a full traversal of the constraint graph may still be required in either case. However, in practice, deeper or broader search may be needed to identify a sufficient variable set, which could pose greater challenges for LLMs.
>
> We expect this increased complexity to result in a drop in LLM accuracy, especially for domains like Logic-Q and Planning-Q, where even 1-sufficient problems already present significant difficulty.
>
> Please see also our response to W1 for reviewer LTGU!

---

### Decision · Program_Chairs · 2025-09-18

**Decision:**

Accept (poster)

**Comment:**

This paper presents a dataset where models are asked to determine what additional piece of information is needed to resolve a reasoning question. The underlying structure of this problem is a CSP, which represents problem underspecification in a formal way (as opposed to semantic ambiguity in natural language).  Three domains are instantiated for QuestBench. Models do well at GSM-derived settings but not planning or logic tasks.

Strengths
1. Timely benchmark addressing a novel aspect of reasoning
2. Carefully constructed dataset to have correct, unique answers for every question
3. Thoroughly evaluated
4. Well-written paper

Weaknesses
1. Multiple-choice format makes the dataset less realistic
2. Limited set of domains (2/4 of which are too easy) that don't necessarily reflect real problems
3. 1-sufficient CSPs are limited in complexity (addressed in rebuttal)

Discussion:
1. The 1-sufficient CSP point has been addressed in my view, through additional theoretical argumentation
2. Trace analysis will be provided in a final version
3. Other points largely stand

Overall, this paper addresses an interesting problem with a solidly-constructed dataset. Although the dataset could be more realistic, more complex, or cover more domains, these strike me as standard limitations that apply to nearly all dataset construction efforts. Personally, I see the lack of open-endedness as the greatest limitation, since it seems that knowing when and how to ask the right questions is a key capability). However, I agree with the authors that this dataset is a careful first step towards solving this problem even if it doesn't achieve everything all at once.